**A multi-tracer approach to constraining artesian groundwater discharge into an**
**alluvial aquifer**
**Authors:** Charlotte P. Iverach[1,2,3], Dioni I. Cendón[1,2,3], Karina T. Meredith[3], Klaus M.
Wilcken[3], Stuart I. Hankin[3], Martin S. Andersen[1,4], Bryce F.J. Kelly[1,2,*]
[1]Connected Waters Initiative Research Centre, UNSW Sydney, NSW, 2052, Australia
[2]School of Biological, Earth and Environmental Sciences, UNSW Sydney, NSW, 2052,
Australia.
[3]Australian Nuclear Science and Technology Organisation, New Illawarra Rd, Lucas
Heights, NSW, 2234, Australia.
[4]School of Civil and Environmental Engineering, UNSW Sydney, NSW, 2052, Australia.
*Corresponding author: bryce.kelly@unsw.edu.au
Understanding pathways of recharge to alluvial aquifers is important for maintaining
sustainable access to groundwater resources. Water balance modelling is often used to
proportion recharge components and guide sustainable groundwater allocations.
However, it is not common practice to use hydrochemical evidence to inform and
constrain these models. Here we compare geochemical versus water balance model
estimates of artesian discharge into an alluvial aquifer, and demonstrate why multi-
tracer geochemical analyses should be used as a critical component of water budget
assessments. We selected a site in Australia where the Great Artesian Basin (GAB), the
largest artesian basin in the world, discharges into the Lower Namoi Alluvium (LNA),
an extensively modelled aquifer, to convey the utility of our approach. Water stable
isotopes ($\delta^{18}O$ and $\delta^2H$) and the concentrations of $Na^+$ and $HCO_3^-$ suggest a continuum
of mixing in the alluvial aquifer between the GAB (artesian component) and surface
recharge, whilst isotopic tracers ($^3H$, $^{14}C$ and $^{36}Cl$) indicate that the alluvial
groundwater is a mixture of groundwaters with residence times of < 70 years and
groundwater that is potentially hundreds of thousands of years old, which is consistent
with that of the GAB. In addition, $Cl^-$ concentrations provide a means to calculate a
percentage estimate of the artesian contribution to the alluvial groundwater. In some
locations, an artesian contribution of up to 70% is evident from the geochemical
analyses, a finding that contrasts previous regional scale water balance modelling
estimates that attributed 22% of all inflow for the corresponding zone within the LNA
to GAB discharge. Our results show that hydrochemical investigations need to be
undertaken as part of developing the conceptual framework of a catchment water
balance model, as they can improve our understanding of recharge pathways and better
constrain artesian discharge to an alluvial aquifer.

## 1 Introduction

Groundwater type mixing in an alluvial aquifer can occur between recently recharged groundwater (via infiltration from the land surface) and groundwater discharging into the alluvium from surrounding geological formations and artesian groundwater resources (Costelloe et al. 2012; Schilling et al. 2016; Rawling & Newton 2016; Salameh et al. 2017). Insufficient spatial and temporal data resolution, as well as heterogeneity in hydrogeological properties can result in considerable uncertainty when proportioning contributions from various sources in groundwater with mixed origins (Anderson & Woessner 1992; Beven 2009; Gardner et al. 2012). Additional uncertainties confounding source attribution include change in the magnitude of groundwater gradients and directions over time due to ongoing groundwater abstraction (for irrigation, stock and domestic water supplies), and the impact that this and flood frequency may have on the extent of artesian discharge and groundwater mixing. These complexities make it challenging to accurately proportion contributions from various sources to an alluvial aquifer and to guide water allocations.

Water balance modelling of alluvial aquifers is commonly used to quantify and proportion recharge inputs from river leakage, floodwaters, areal (diffuse recharge) and discharge from artesian sources (Anderson & Woessner 1992; Middlemis et al. 2000; Zhang et al. 2002; Dawes et al. 2004; Barnett et al. 2012; Giambastiani et al. 2012; Hocking & Kelly 2016). Historically, hydrochemical analyses are not often used to constrain catchment scale water balance modelling (Reilly and Harbaugh 2004; Barnett et al. 2012), despite Scanlon et al. (2004) highlighting the need to use multiple techniques (including hydrochemical insights) to increase the reliability of recharge and discharge estimates. Geochemical data can improve our understanding of groundwater mixing processes because of the potential to trace pathways of groundwater movement and water-rock interactions, whilst also providing insights on the impacts of past groundwater extractions (Edmunds 2009; Martinez et al.

2017). Therefore, the integration of geochemical evidence to constrain aquifer water balance
models provides a more rigorous approach for proportioning input sources for groundwater
that has mixed origins (Raiber et al. 2015; Currell et al. 2017).

Radioactive isotopic tracers that provide insights into groundwater residence times can

constrain mechanisms of recharge and discharge, and detect groundwater mixing. Isotopes of
dissolved species can be useful for elucidating groundwater mixing provided the different
sources of groundwater have distinctly different and consistent isotopic signatures. However,
each tracer has a different half-life and both physical and chemical processes and calculation
assumptions can affect the interpretation of groundwater residence times (Jasechko 2016).
Therefore, multiple tracers are useful for covering the relevant time scales and uncertainties
associated with the large range of groundwater residence times. Tritium ($^{3}$H) is an excellent
indicator of modern recharge inputs in shallow groundwater (Robertson et al. 1989; Chen et
al. 2006; Duvert et al. 2016), and provides valuable information on processes active in the
past ~ 70 years. Carbon-14 ($^{14}$C) is used to understand processes active from modern to ~ 30
ka (Clark & Fritz 1997; Cartwright et al. 2010; Cendón et al. 2014) and chlorine-36 ($^{36}$Cl),
whilst applicable in modern groundwater (Tosaki et al. 2007), is usually reserved for the
identification of much older groundwater (100 ka to 1 Ma). One of the challenges of using
$^{36}$Cl is that, in certain cases, nucleogenic production of $^{36}$Cl can be significant and/or varying
Cl concentrations can complicate groundwater residence time interpretations. Additionally,
the interpretation of $^{36}$Cl can be affected by the input function, as $^{36}$Cl values from rainfall
vary temporally. This means that the input function for rainfall from any time in the past may
be different from current conditions (Phillips 2000). However, in regions with low and fairly
consistent Cl concentrations (such as in our study area), $^{36}$Cl values can provide solid
indications of old groundwater residence times (Mahara et al. 2007).

These isotopes can also be used for tracer mixing calculations independent of residence

time estimations (Bentley et al. 1986; Andrews & Fontes 1993; Love et al. 2000; Moya et al.

2016). Therefore, the combination of $^{3}$H, $^{14}$C, and $^{36}$Cl dating techniques can provide

hydrochemical process insights that cannot be captured by using only one isotope.

Identification of recharge and discharge pathways (particularly from underlying

artesian contributions), and proportioning their relative contributions in a groundwater

sample can be better constrained by combining traditional geochemical data with multiple

dating techniques and other hydrologic analyses (Amiri et al. 2016; Rawling & Newton 2016;

Schilling et al. 2016). This is because groundwater geochemical data give insights into long-

term patterns of mixing and groundwater flow, whereas other hydrologic data (such as

hydraulic head differences) provide insights on seasonal pumping impacts, and current local

and catchment-scale groundwater flow paths.

Here, we present for the first time a multi-tracer approach to constraining artesian

discharge from the Great Artesian Basin (GAB) into the Lower Namoi Alluvium (LNA),

north-west New South Wales (NSW), Australia (Figure 1). We use water stable isotopes and

major ion data to assess the major recharge and discharge pathways and occurrences of

groundwater mixing in the LNA. We also use $^{3}$H, $^{14}$C and $^{36}$Cl to show that artesian discharge

from the underlying GAB to the LNA is locally much higher than is currently estimated from

water balance models used to guide groundwater allocations in the region (Lower Namoi

Groundwater 2008). Our results highlight the need to consider a multi-tracer geochemical

approach when assessing artesian contributions to alluvial aquifers and constraining water

balance models of alluvial systems globally.

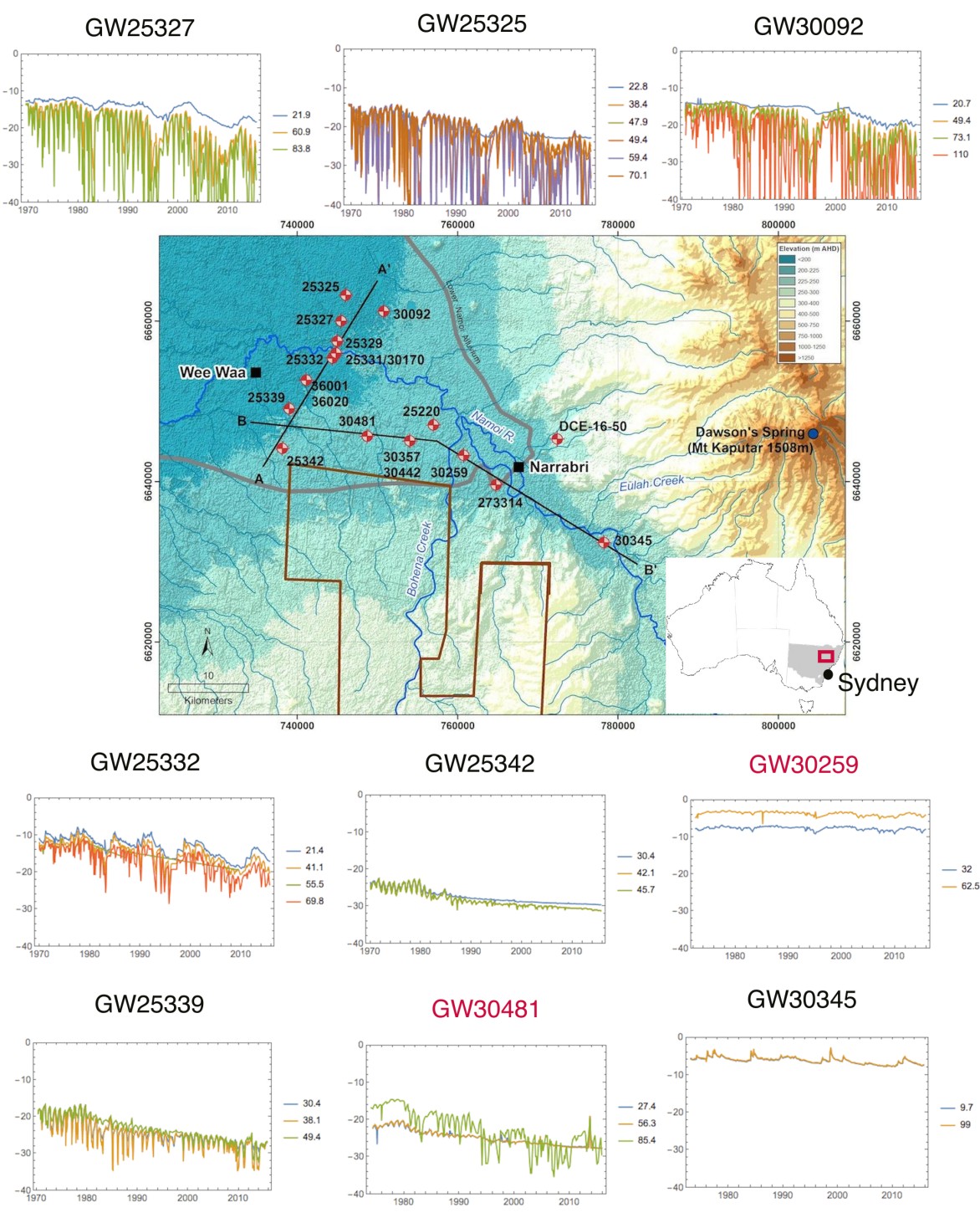

**Figure 1.** Map of the study area and sample locations, along with the location of the study area in Australia. Accompanying hydrographs show the groundwater level response in different piezometers throughout the study area (groundwater level data sourced from BOM 2017). The different colours in the hydrographs represent the different monitoring bores in the nested set. The bottom of the slotted

interval for each bore is shown in the key. The x-axis in each hydrograph is the year (1970-2010) and
the y-axis is depth (between 0 and 40 m below ground surface (bgs)). The two locations with red text
highlight areas where the hydrograph heads show clear GAB contribution, with the deeper piezometer
showing a higher head than the shallow one. The remaining locations show no apparent GAB
contribution to the LNA based on the hydrograph data.

**2      Study Area**
The lower Namoi River catchment is located in the north-west of NSW, Australia (Figure 1).
Groundwater resources in the LNA are the most intensively developed in NSW (DPI Water
2017). For this reason, there is concern regarding groundwater exploitation and threat to the
long-term sustainability of the system (Lower Namoi Groundwater 2008; DPI Water 2017).
Groundwater abstraction from the LNA supports a multibillion-dollar agricultural sector
(focused around cotton growing established in the 1960s), supplying around 50% of water for
irrigation in the region (Powell et al. 2011). Peak extraction of approximately 170 x $10^6$ m$^3$
occurred over the 1994/1995 growing season (Smithson 2009). Consistently declining
groundwater levels and concern regarding the long-term sustainability of groundwater
abstraction led to the implementation of a Water Sharing Plan in 2006, which systematically
reduced groundwater allocations to the irrigation sector over a ten-year period. The present
allocation is 86 x $10^6$ m$^3$ /year (Lower Namoi Groundwater 2008).

**2.1      Hydrogeological setting**
The lower Namoi River catchment lies within the Murray-Darling Basin, overlying the
Coonamble Embayment, which is in the south-east portion of the GAB (Radke et al. 2000).
The southernmost portion of the LNA is underlain by Triassic formations, while northwest of
monitoring bore 30345 the LNA is underlain by Jurassic formations (Figure 2). Within the
region of study, the oldest outcropping bedrock formation is the early Triassic Digby
Formation (lithic and quartz conglomerates, sandstones and minor finer grained sediments)
(Tadros 1993). The Digby Formation outcrops in the south-east of the area and the Namoi
River abuts the formation just south of B' on Figure 2. The Digby Formation is overlain by
the Triassic Napperby Formation (thinly bedded claystone, siltstones and sandstone). This
formation occurs at a depth of 106 m, just below the base of monitoring bore 30345 (NSW
Pinneena Groundwater Database, driller logs).In outcrops to the east of the study area, the
Napperby Formation is overlain by the late Triassic Deriah Formation (green lithic sandstone
rich in volcanic fragments and mud clasts) (Tadros 1993). The boundary between the Triassic
and Jurassic lies west of monitoring bore 30345. The Jurassic formations important to this
study are the Purlawaugh Formation (carbonaceous claystone, siltstone, sandstone and
subordinate coal), Pilliga Sandstone (medium to coarse quartzose sandstone) and the Orallo
Formation (clayey to quartzose sandstone, subordinate siltstone and conglomerate) (Tadros
1993). The Pilliga Sandstone forms the bedrock below monitoring bores 25325 to 25342, and
in the Namoi region is the primary aquifer of the GAB.

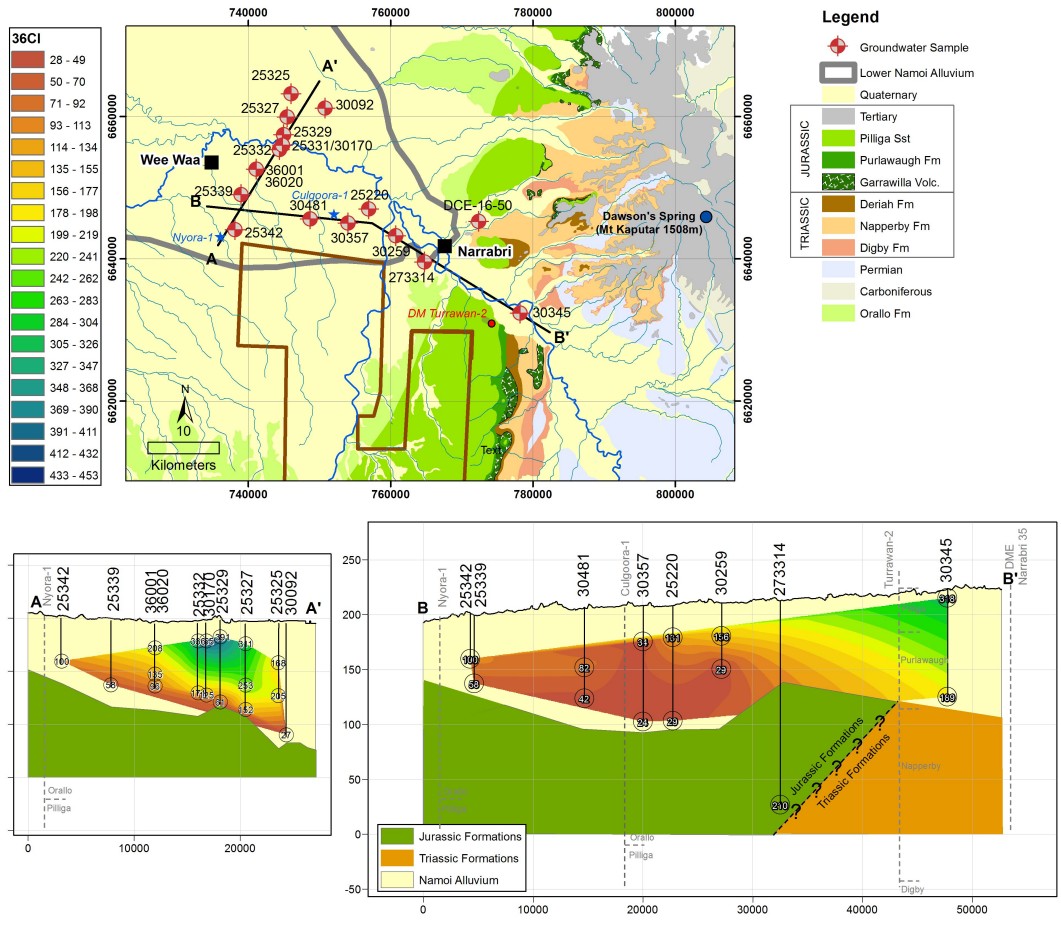

**Figure 2.** A geological map of the study area and two cross sections through the study area, showing the location and depth of the samples in the alluvium and their proximity to formations of the GAB. Contacts obtained from gas wells Nyora-1, Culgoora-1 and Turrawan-2, coinciding with our cross sections, are added. Their locations are displayed on the map. The general direction of groundwater flow is from SE to NW, aligning with the B-B' line on the map. The chlorine-36 data interpolated using the 'natural neighbours' algorithm is shown in each cross section.

From the late Cretaceous to the mid Miocene, a palaeovalley was carved through the basement rocks (Kelly et al. 2014). Then from the mid Miocene until present, the palaeovalley was filled with reworked alluvial sediments. Groundwater abstraction in the study area is mostly from these alluvial sediments. Fluvial and aeolian interbedded clays, silts, sands and

gravels form the up to ~ 140 m thick alluvial sequence of the Lower Namoi Catchment
(Williams et al. 1989). Traditionally, three main non-formally defined aquifers/formations have
been used to describe the LNA. The semi-confined Cubbaroo Formation overlies the bedrock in
the northern palaeochannel (which passes beneath monitoring bores 25325 and 30092). This
formation is up to 60 m thick. The Cubbaroo Formation is overlain by the semi-confined
Gunnedah Formation, which is up to 80 m thick, and is conformably overlain by the unconfined
Narrabri Formation, which is 10 to 40 m thick (Williams et al. 1989). However, recent studies
in the Namoi Catchment suggest that the rigid subdivision in to the Narrabri, Gunnedah, and
Cubbaroo formations cannot easily explain the continuum in chemical evolution observed
(discussed further below) and that the valley filling sequence is better characterised as a
distributive fluvial system (Kelly et al. 2014, Acworth et al. 2015).

Groundwater drains from the Upper Namoi into the LNA via a bedrock constriction north

of Narrabri and generally flows from east to west within the LNA (Barrett 2012). Hydraulic
conductivity in the alluvial aquifer is highly variable (0.008-31 m/day) due to the presence of
variable sand and clay (Golder Associates 2010). However, hydraulic conductivity generally
increases with depth.

**2.2    Current understanding of recharge and discharge processes in the Lower Namoi**
**Alluvium**
There have been numerous catchment water balance models and hydrochemical
investigations in the study area because of the local and national economic importance of the
LNA. However, the hydrochemistry of the groundwater in the region has not been used in
conjunction with water balance modelling prior to this study (Merrick 2000; CSIRO 2007;
Kelly et al. 2007).

*2.2.1   Water balance modelling of recharge*
To guide groundwater allocations from the LNA, a series of water budget models were
developed using MODFLOW (Merrick 2000; summarised in Kelly et al. 2007). These
models were driven by climatic, rainfall, flood and streamflow data and calibrated to
groundwater head data. There are multiple plausible solutions for all water balance models
and the solution presented is often constrained by several factors. These constraining factors
include geological insights; the modeller's experience and biases (such as, for example, the
way diffuse recharge is modelled either as a percentage of rainfall (Merrick 2000; CSIRO
2007) or as a complex evapotranspiration function (Giambastiani et al. 2012)); verification
measures and pragmatic goals. One MODFLOW derived water balance model proportioned
the recharge for the water budget period 1980-1994 as following: flood and diffuse rain
recharge 24.1 x $10^5$ m$^3$ /year, stream recharge 33.7 x $10^5$ m$^3$ /year, up gradient alluvial inflow
3.06 x $10^5$ m$^3$ /year, and artesian (GAB) recharge 9.5 x $10^5$ m$^3$ /year.  In that model, artesian
recharge was inferred to occur in the eastern portion of the model (between Narrabri and Wee
Waa), which overlaps with this study area (Figure 1). The zone between Narrabri and Wee
Waa accounted for 42.7 x $10^5$ m$^3$ /year of the total recharge to the LNA. Thus, according to
the model, GAB discharge into the LNA in this area equated to 22%. When the LNA
MODFLOW model was calibrated there was no consideration given to using hydrochemical
data to constrain the calibration (Merrick 2000; CSIRO 2007; Kelly et al. 2007).

*2.2.2   Hydrochemical estimates of recharge*
The first isotopic investigation in the area was conducted from 1968 to 1975 and partially
published by Calf (1978). The author used $^{14}$C and $^3$H to assess recharge pathways to the
LNA and found evidence for river recharge in the upper aquifer, and that modern
groundwater penetrated the deeper parts of the LNA. Calf (1978) also found evidence for
'leakage' of groundwater from the GAB up into the deeper LNA, however volumetric
estimates were not provided.
McLean (2003) conducted an extensive hydrochemical and isotopic characterisation of
both the GAB groundwater and the alluvial groundwater in 1999-2000. This research
concluded that mixing of groundwater from the GAB into the lower and middle parts of the
LNA is an important process especially in the south of the catchment. This study also did not
quantify the amount of mixing occurring between the two groundwater sources.
The over-reliance of water balance models used to allocate groundwater resources that
have not been constrained by isotopic tracer residence times or hydrochemical results is a
common issue globally. This research highlights that hydrochemical investigations improve
our conceptual understanding of recharge pathways and that such investigations should be
applied to all important groundwater resource assessments to enable sustainable management.

**3      Materials and methods**
**3.1      Groundwater collection**
This study comprised two field campaigns, the first one from 28 January 2016 to 8 February
2016 (summer) when the aquifer was stressed by pumping for irrigation, and the second from
21 June 2016 to 30 June 2016 (winter) in the absence of abstraction for irrigation.
In summer, 28 groundwater samples were collected from NSW Department of Primary
Industries Water (DPI Water) monitoring bores and a surface water sample from the Namoi
River. In winter, 16 groundwater samples were collected from NSW DPI Water monitoring
bores and surface water samples from the Namoi River and 2 upstream tributaries (see
Supplementary Table 2 for locations). The bores are screened at varying intervals (average
length of screened interval: 5.6 m (see Supplementary Table 2 for individual bores)),
intersecting the shallow, middle and deep alluvium. Most bores were sampled with either a
Grundfos (MP1 sampling pump) or Bennett compress air piston pump, with the pump placed ~
1 m above the screen when using the Grundfos pump. Drop-tube extensions were used with the
Bennett pump to place the pump intake just above the screen. Some deep monitoring bores
were sampled with a portable bladder pump using low-flow methods (Puls & Barcelona, 1996).
In these bores the pump was placed approximately 10 m below standing water level, with a
drop-tube cut to place the pump intake within the screen. For shallower bores (less than 50 m),
a 12 V battery operated pump was used with the pump intake placed ~1 m above the screen.
For all sample sites, physico-chemical parameters (pH, DO, EC) were monitored and samples
collected once three well volumes had been pumped and/or the physico-chemical parameters
stabilised. This was generally achieved within 1 to 3 hours after onset of pumping. Sample
collection involved an in-line, 0.45 μm, high-volume filter connected to a high-density
polyethylene (HDPE) tube. Total alkalinity concentrations (field alkalinity) were determined in
the field by acid titration using a HACH digital titrator and external pH meter control. The $Fe^{2+}$
and $HS^-$ concentrations were determined using a portable colorimeter (HACH DR/890).

Samples for anion and water stable isotope ($\delta^2H$ and $\delta^{18}O$) analyses were collected in 60

mL and 30 mL HDPE bottles, respectively, with no further treatment. Samples for cation
analysis were collected in 60 mL HDPE bottles and acidified with ultrapure nitric acid.
Samples for $^{14}C$ and $^3H$ were collected in 1 L narrow mouth HDPE bottles and 2 L HDPE
bottles respectively, and were sealed with tape to avoid potential atmospheric exchange during
storage. Samples for $^{36}Cl$ were collected in 1 L narrow mouth HDPE bottles with no further
treatment. Major ion and $^{14}C$ samples were refrigerated at $4^oC$ until analysed.

We were not able to access any previously sampled GAB bores within the study area.

Thus, better constrain GAB groundwater characteristics, we used geochemical data from
known GAB bores collected by Radke et al. (2000) and McLean (2003). These data were
collected to the northwest of our study area and are used as a range (depending on availability
of the original reported data) for the GAB end-member in our discussions (Supplementary
Table 1).
To help in the description of results, we use shallow (< 30 m), intermediate (30 − 80 m)
and deep (> 80 m) as a rough guide to the origin of the groundwater sample. The chosen depth
categories are based on clusters and trends in the $^{14}C$ analyses.

**3.2 Geochemical analyses**
Groundwater samples from both campaigns were analysed at ANSTO by inductively coupled
plasma atomic emission spectroscopy (ICP-AES) for cations and ion chromatography (IC) for
anions. Samples for $\delta^2H$ and $\delta^{18}O$ were analysed using Cavity Ring-Down Spectroscopy
(CRDS) on a Picarro L2130-*i* analyser. These values are reported as ‰ deviations from the
international standard V-SMOW (Vienna Standard Mean Ocean Water) and results have a
precision of ± 1‰ for $\delta^2H$ and ± 0.15‰ for $\delta^{18}O$.
The $^{14}C$ samples were processed and analysed at ANSTO using methods described in
Cendón et al. (2014). The $^{14}C$ activities were measured by accelerator mass spectrometry
(AMS) using the ANSTO 2MV tandetron accelerator, STAR (Fink et al. 2004). The $^{14}C$ results
were reported as percent modern carbon (pmc) following groundwater $^{14}C$ reporting criteria
(Mook & van der Plicht 1999; Plummer & Glynn 2013) with an average 1σ error of 0.21 pmc.
The $^3H$ samples were analysed at ANSTO. Water samples were distilled and
electrolytically enriched prior to analysis by liquid scintillation. The $^3H$ concentrations were
expressed in tritium units (TU) with a combined standard uncertainty of ± 0.03 TU and
quantification limit of 0.04 TU. Tritium was measured by counting beta decay in a liquid
scintillation counter (LSC). A 10 mL sample aliquot was mixed with the scintillation cocktail
that releases a photon when struck by a beta particle. Photomultiplier tubes in the counter
convert the photons to electrical pulses that are counted over 51 cycles for 20 minutes.

The $^{36}Cl/Cl$ and $^{36}Cl/^{37}Cl$ ratios were measured by AMS using the ANSTO 6MV SIRIUS

Tandem Accelerator (Wilcken et al. 2017). Samples were processed in batches of 10, with
each batch containing 1 chemistry blank. The amount of sample used was selected to yield ~
5 mg of Cl for analysis without carrier addition. Chloride was recovered from the sample
solutions by precipitation of AgCl from hot solution (Stone et al. 1996). This AgCl was re-
dissolved in aqueous $NH_3$ (20-22 wt %, IQ grade, Seastar) to remove sulfur compounds of Ag.
Owing to isobaric interference of $^{36}S$ with $^{36}Cl$ in the AMS measurements, a saturated
$Ba(NO_3)_2$ solution (99.999% trace metal basis) was used to precipitate sulfur as $BaSO_4$. At
least 72 h were allowed for $BaSO_4$ to settle from a cold solution (4℃) in the dark before
removal of the supernatant by pipetting and filtration (0.22 Millex GS). Pure AgCl was re-
precipitated by acidifying the $Ag(NH_3)_2$-Cl solution with 5M nitric acid (IQ Seastar, sub-
boiled). Finally, AgCl was recovered, washed twice and dried. It was then pressed into high-
purity AgBr (99% trace metal basis, Aldrich) in 6 mm diameter Cu-target holders. AgBr has a
much lower sulfur content than Cu. The stable Cl isotopes $^{35}Cl$ and $^{37}Cl$ were measured with
Faraday cups and $^{36}Cl$ events were counted with a multi-anode gas ionisation chamber. Gas
(Ar) stripping (for good brightness/low ion straggling) the ions to 5+ charge state in the
accelerator terminal suffices for effective $^{36}S$ interference separation in the ionisation chamber
combined with sample-efficient and rapid analysis. Purdue PRIMELab Z93-0005 (nominally
$1.20 \times 10^{-12}$ $^{36}Cl/Cl$) was used for normalisation with a secondary standard (nominally $5.0 \times 10^{-13}$
$^{36}Cl/Cl$ (Sharma et al. 1990)) used for monitoring. Background subtraction was done with a
linear dependence between $^{36}Cl$-rate and interfering $^{36}S$-rate. This dependency is established by
combining all the blank and test sample measurements and applied to the unknown samples
during offline data analysis. This correction factor was typically less than analytical uncertainty
of 3-4% bar one sample that had a correction factor of 12% with an analytical uncertainty of

6%.


## 3.3     Geochemical calculations


Calculations necessary to assess electrical neutrality, dissolved element speciation and
saturation indices for common mineral phases were undertaken using the PHREEQC
Interactive program (3.3.8) (Parkhurst & Appelo 1999) and the incorporated WATEQ4F
thermodynamic database (Ball & Nordstrom 1991). The cation and anion analyses were
assessed for accuracy by evaluating the charge balance error percentage (CBE%). All samples
fell within the acceptable ± 5% range, except for samples 25327-1 (-7.8 %) and 36001-1 (-5.8
%). The inverse geochemical modelling code NEPATH XL (Plummer et al. 1994; Parkhurst &
Charlton 2008) has been used to calculate the mixing ratio between two end-members, using
their Cl concentrations. The choice of end-members will influence calculated proportions,
however, end-members were selected to provide conservative approximations.
Despite limitations, $^{36}Cl$ residence times for selected low $^{36}Cl/Cl$ samples were
calculated from the equations of Bentley et al. (1986). This allows a direct comparison, under
similar assumptions, with other estimates obtained from GAB groundwater elsewhere
(Bentley et al. 1986; Radke et al. 2000; Love et al. 2000; Moya et al. 2016) and within the
Coonamble Embayment (Radke et al. 2000; Mahara et al. 2007). These calculations assume a
piston flow setting with no other sources or sinks besides recharge and natural decay (eqn. 1):
$$t = \frac{-1}{\lambda_{36}} \ln \frac{R - R_{se}}{R_0 - R_{se}} \qquad (1)$$
where R = $^{36}Cl/Cl$ ratio measured in the sample, $R_0$ = the initial $^{36}Cl/Cl$ ratio (meteoric
water), $R_{se}$ = the $^{36}Cl/Cl$ ratio under secular equilibrium (in this case the $^{36}Cl/Cl$ ratio from the
Pilliga Sandstone), and $\lambda_{36}$ is the decay constant (2.303 x $10^{-6}$). We used a $R_0$ value of 160
$(x10^{-15})$, which was an average of 10 samples compiled from studies in the Coonamble
Embayment and reported in Radke et al. (2000). For $R_{se}$ a value of 5.7 $(x10^{-15})$ was used,
which is appropriate for aquifers dominated by sandstone (this secular equilibrium value can
vary according to the dominant lithology). This $R_{se}$ value has been applied to $^{36}$Cl/Cl
calculations elsewhere in the GAB (Moya et al. 2016) and is similar to that calculated from
drill-core samples recovered in the GAB by Mahara et al. (2009).

**4      Results**
**4.1      Major ion chemistry**
The groundwater of the alluvial aquifer is predominantly Na-HCO$_3$-type water, with
concentrations ranging from 0.12 mmol/L to 54.6 mmol/L (average: 6.85 mmol/L; std dev:
8.7 mmol/L) for Na$^+$ and 0.29 mmol/L to 24.0 mmol/L (average: 6.43 mmol/L; std dev: 4.8
mmol/L) for HCO$_3^-$ (Supplementary Table 2). Generally, the highest concentrations of Na$^+$
and HCO$_3^-$ occur in the deeper groundwater and decrease up the vertical groundwater profile
(Figure 3a). The concentration of these two ions in the groundwater of the LNA is higher
than expected from local rainfall sources and other shallow groundwater alluvial systems in
eastern Australia (Martinez et al. 2017). In GAB groundwater, the Na-HCO$_3$ molar ratio is
generally 1:1 and the two ions are generally present in higher concentrations than in our
alluvial samples (Radke et al. 2000; McLean 2003), which is evident in the position of the
regional GAB samples in Figure 3a.
Additional ions used in this study are F$^-$, Cl$^-$ and the Cl/Br ratio. The concentration of F$^-$
in the groundwater ranges from 0.002 mmol/L to 0.215 mmol/L (average: 0.028 mmol/L; std
dev: 0.04 mmol/L). Fluoride concentrations generally increase with depth and accumulate in
solution as all groundwater samples are below saturation with respect to fluorite (Figure 3b).
Concentrations of Cl$^-$ in the alluvial groundwater range from 0.063 mmol/L to 26.73 mmol/L
(average: 1.67 mmol/L; std dev: 3.7 mmol/L). Unlike the other major ions, Cl$^-$ concentrations
through the vertical groundwater profile are relatively stable (Figure 3c). The relationship
between Cl$^-$ and the Cl/Br ratio shows that groundwater composition clusters from values
below the seawater ratio to values close to seawater. The Cl/Br ratios are similar to ranges
found in other alluvial groundwater systems but slightly lower than ratios observed in other
GAB samples for Australian locations (Herczeg et al., 1991; Cendón et al., 2010; Cartwright
et al., 2010). Additionally, the Cl/Br ratios in shallow samples connected to the river are
consistent with expected ratios in rainfall (Short et al. 2017). The regional GAB samples
(Radke et al. 2000) show a Cl/Br ratio more similar to seawater, with our samples from the
LNA lying on a mixing trend between the two end-members (Figure 3c).

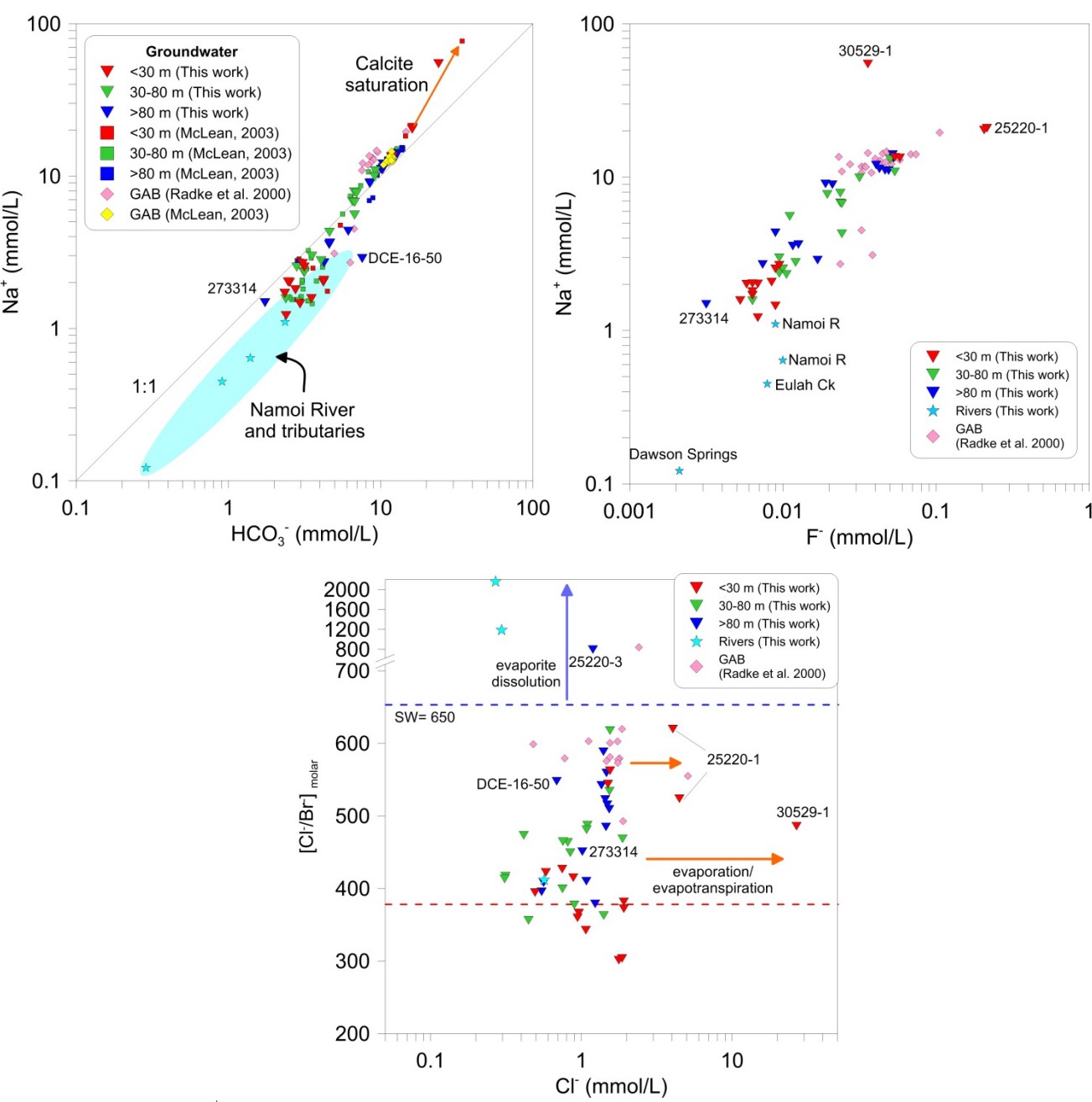

**Figure 3.** a) Na$^+$ vs HCO$_3^-$ showing the mixing trend that the alluvial samples form between the Namoi River and samples from the GAB (Radke et al. 2000; McLean 2003). The orange calcite saturation line indicates samples that are more enriched due to separate evapotranspiration and calcite precipitation. The shaded blue ellipse represents all river chemistry data available for the Namoi River and tributaries (this work (n=4), McLean 2003 (n=4), Mawhinney 2011 (n=79)); b) Na$^+$ vs F$^-$ and c) Cl$^-$/Br$^-$ vs Cl$^-$, highlighting the mixing trend between the surface recharge and the GAB that we observe in other geochemical indicators. The red dotted line represents the Cl$^-$/Br$^-$ ratio for rainfall and the blue dotted line is the seawater ratio.

We identified one major outlier in the hydrochemical results, which was sample
273314. This sample is from 207 m bgs and the bore screen is classified as being in the GAB.
However, the geochemical parameters for this deep GAB sample have a signature more
similar to river water than what would be expected in the GAB 207 m bgs. The concentration
of $Na^+$, $HCO_3^-$, $Cl^-$, $F^-$ and the Cl/Br ratio in this sample plot closer to the river and shallow
groundwater than the deeper groundwater system (Figure 3). Potential reasons for this are
explored in detail below.

**4.2      Stable water isotopes ($\delta^2$H and $\delta^{18}$O)**
The stable water isotopic values for this study range from -0.76‰ to 8.4‰ for $\delta^{18}$O and -
7.5‰ to -54.9‰ for $\delta^2$H. Most groundwater samples cluster together at around -6‰ and -
40‰ ($\delta^{18}$O and $\delta^2$H) and lie on the global meteoric water line (GMWL), to the right of the
nearest available local meteoric water lines (LMWL) (Macquarie Marshes and Gunnedah)
(Figure 4; Supplementary Table 3). A group of mostly shallow samples collected from
piezometers close to river channels define a trend to the right of the GMWL with a slope of
5.96, which is consistent with evaporation (Cendón et al. 2014). Our results are similar,
including the shallow groundwater evaporative trend, to those recorded by McLean (2003).
Water stable isotopic compositions for regional GAB samples range from -6.58‰ to -6.24‰
for $\delta^{18}$O and -43.1‰ to -38.8‰ for $\delta^2$H (McLean 2003) (Figure 4).

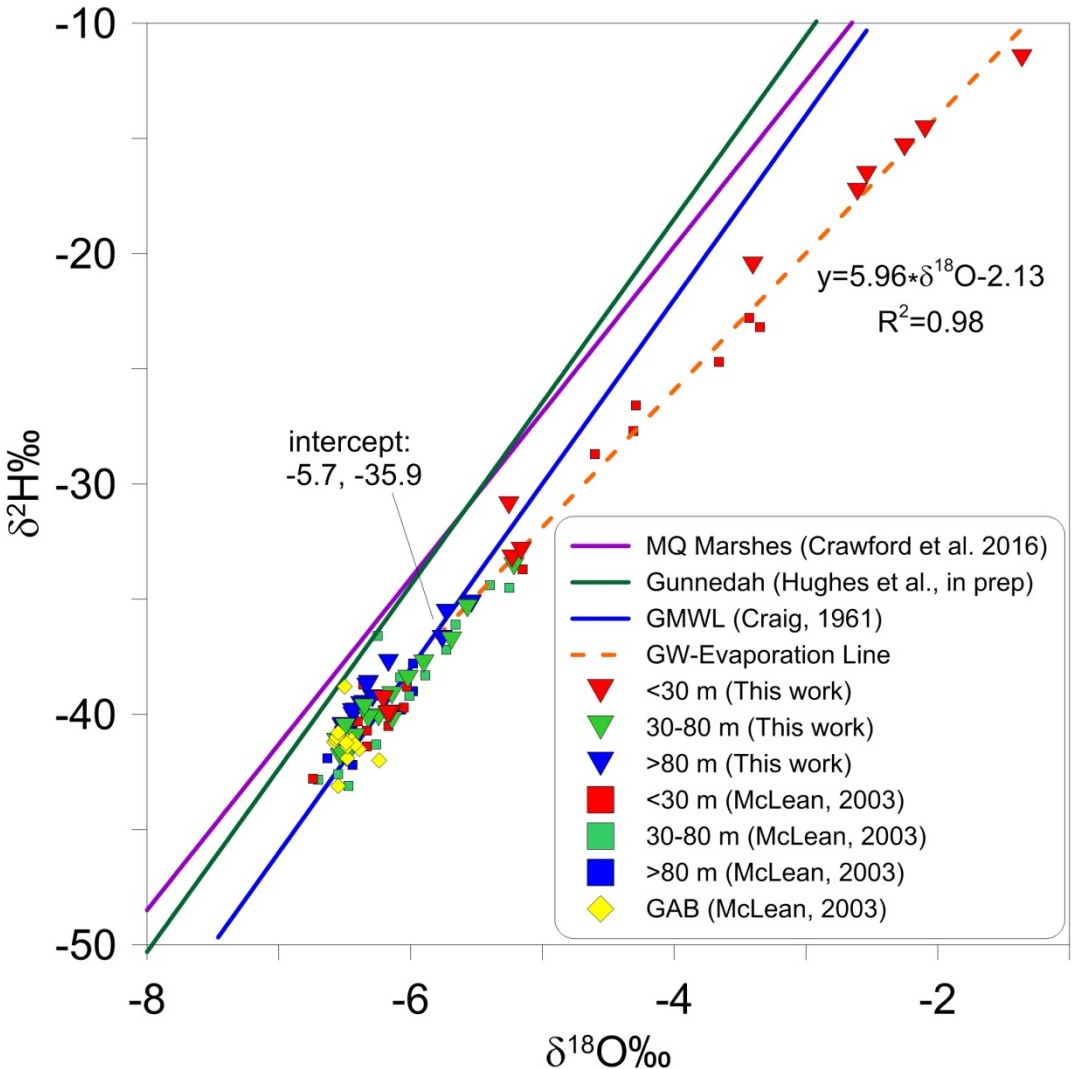

**Figure 4.** Water stable isotopes in the LNA, showing the two separate mechanisms of recharge; surface water recharge plotting along an evaporation trend line and potential inflow from the GAB clustered with regional samples from the GAB (McLean 2003).

## 4.3 Isotopic tracers ($^3$H, $^{14}$C and $^{36}$Cl)

Tritium activities vary throughout the study area, ranging from below the quantification limit (< 0.04 TU) to 2.36 TU (average: 0.42 TU). Tritium activities generally decrease with depth and distance from the river channel (Figure 5) (all data in Supplementary Table 3). The highest $^3$H activities of 2.31 TU and 2.36 TU are from a sample 40 m from the river and the

Namoi River itself, respectively. These are very similar to modern rainfall in Australia (~ 2-3
TU (Tadros et al. 2014)), which suggests modern recharge near the river channels However,
$^3$H > 0.04 TU was measured at depth (down to 207 m bgs). The $^3$H activities we measured at
depth are significant for Australian groundwater, as the peak of the bomb pulse in Australia
was only around 60 TU compared to locations in the northern hemisphere. This is primarily
because most thermonuclear testing was undertaken in the northern hemisphere far from
Australia and mixing is limited between the atmospheric convection cells in the northern and
southern hemispheres. Therefore, $^3$H in Australian rainfall has been at natural background
concentrations for some time (Tadros et al. 2014).

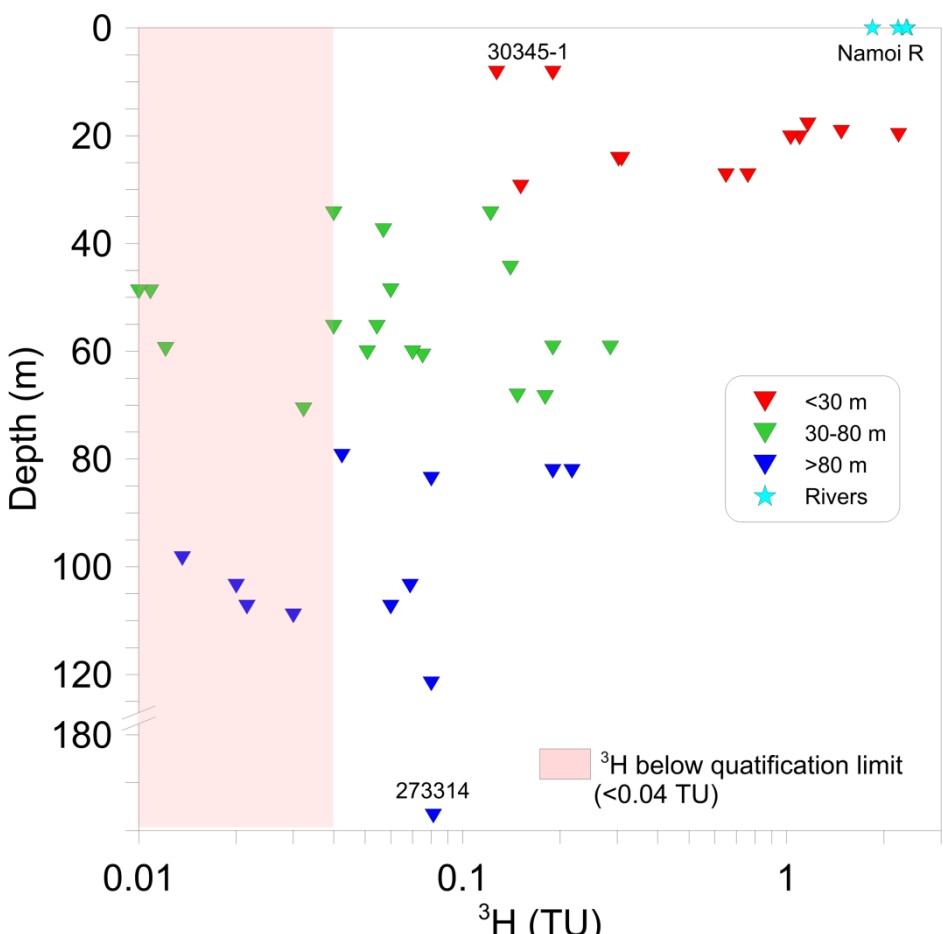

**Figure 5.** Plot of depth vs $^3$H, highlighting the $^3$H activity throughout the vertical groundwater profile.
Samples that fall within the pink zone on the left are below the quantification limit (< 0.04 TU). These
data are not included in our interpretation of how $^3H$ changes with depth. They are presented to
convey the relative proportions of interpretable versus non-interpretable data.
The $^{14}C$ content in the groundwater ranged from 0.2 pmc to 107.6 pmc (average: 54.0
pmc). Generally, groundwater samples shallower than 30 m had a high $^{14}C$ content (> 90
pmc), which decreased with depth. There were 9 samples with a $^{14}C$ content below 1 pmc,
indicating old groundwater (> 30 ka), with total depths ranging from 35 m bgs to 207 m bgs.
Our $^{36}Cl$ results for the alluvial groundwater ranged from 24.06 ($\times10^{-15}$) to 455.35 ($\times10^{-15}$)
($\times10^{-15}$) (average: 169.4 ($\times10^{-15}$) (shown in the interpolation in Figure 2). It has been found that
groundwater in the GAB recharge zone closest to the study area has a $^{36}Cl/Cl$ ratio up to ~
200 ($\times10^{-15}$) (Radke et al. 2000) with recharge values applied in calculations elsewhere in the
GAB of 110 ($\times10^{-15}$) (Moya et al. 2016). Water from the Namoi River has a $^{36}Cl/Cl$ ratio of ~
420 ($\times10^{-15}$) (Supplementary Table 4).

**5      Discussion**
**5.1      Identification of recharge and mixing between the GAB and the LNA**
The $\delta^{18}O$ and $\delta^2H$ isotopic compositions suggest two mechanisms of recharge to the
alluvium: artesian discharge and surface water infiltration. The regional GAB samples plot
within the alluvial groundwater sample range, suggesting a GAB component in the alluvium.
The evaporation line in Figure 4 indicates recharge to the alluvium via surface water
infiltration. It also shows a good connection between surface water that has undergone
evaporation prior to recharge.
Additional evidence for these two mechanisms of recharge is the composition of $Na^+$
and $HCO_3^-$ in the LNA. Figure 3a shows a mixing line that the alluvial samples follow,
plotting between the end-members of the GAB and the Namoi River, suggesting an
increasing GAB contribution to the alluvial groundwater with depth. This also implies that a
continuum of mixing exists between the shallow and deep groundwater within the LNA. The
shallow samples (25220-1 and 30259-1) that are more $Na^+$ enriched compared to samples
from the GAB have undergone separate evapotranspiration processes and hence have a
concurrent increase in $Cl^-$. Assuming that $Cl^-$ is behaving conservatively (Appelo & Postma
2005) we surmise that increases in dissolved major ion concentrations concomitant with
increases in $Cl^-$ in the shallow groundwater are likely to be a result of evapoconcentration.

Further hydrochemical evidence for these recharge mechanisms is the covariation of

$Na^+$ and $F^-$, both interpreted as primarily derived from groundwater interaction with silicate
minerals in this region (Airey et al. 1978; Herczeg et al. 1991; McLean 2003) (Figure 3b).
Our alluvial samples fall on the mixing line between samples from the river and nearby
tributaries and regional samples from the GAB (Radke et al. 2000), in a similar way to the
$Na-HCO_3$ trend that we observe in Figure 3a. The Cl/Br ratios in the groundwater also
support the mixing interpretation provided by the $Na^+$ and $HCO_3^-$ concentrations, contrary to
the possibility of water rock interactions along the alluvium flowpath (Figure 3c).
Furthermore, the relationship between $^{36}Cl$ and $Na^+$ provides additional evidence of mixing in
the groundwater (Supplementary Figure 1).

Figure 3 also highlights the deep outlying sample (273314), which was 207 m bgs in

total depth, yet plots with the shallow alluvial and river samples. Figure 2 shows that this
sample is situated just above the Napperby Formation. We hypothesise that this sample
originated from surface recharge from the Namoi River (which is in contact with the
underlying Digby Formation to the south of the study area), with negligible input from the
more $Na-HCO_3$-rich groundwater in the Pilliga Sandstone, where the sample is from. Sample
30345-2 (Supplementary Tables 2 and 3), which is situated in the lower part of the LNA in
proximity to the alluvial contact with the Napperby Formation (Figure 2) has a similar
geochemistry. These results suggest the connection between deeper Triassic formations
beneath the GAB and the Namoi River, which must be an important consideration in future
water balance models of the catchment.

*5.1.1 Mixing between groundwaters of varying residence times*
Major ion and water stable isotope data suggest two primary mechanisms of recharge to
the LNA and show that mixing is occurring within the alluvium. $^3$H activity and $^{14}$C content
in the alluvial groundwater to quantify the potential residence times of the groundwater
sources that are mixing within the alluvium. Tritium activities > 0.04 TU at depth (down to
207 m bgs) indicates the extent of recharge from episodic flooding. Measuring $^3$H > 0.04 TU
at these depths also shows that surface recharge reaches the deeper LNA relatively quickly (<
70 years). Tritium data from the 1970's collected from bores that were included in our
sampling campaign (25329 and 25332) (Calf 1978) suggest that $^3$H was already present in the
deeper parts of the alluvial aquifer (> 70 m bgs) prior to a major flood in 1971, with activities
ranging from 7.9 TU to 11.2 TU. This indicates good connectivity to and recharge from the
surface. Additionally, measurements of $^3$H in these bores post-flooding (16.6 to 20.7 TU)
indicate that substantial recharge from the surface took place during this flood. This
highlights the importance of surface water recharge to the LNA. The activities of $^3$H > 0.04
TU throughout the vertical profile of the LNA (Figure 5) are inconsistent with the low $^{14}$C
contents in the groundwater. The presence of measurable $^3$H but negligible $^{14}$C (close to 0
pmc) suggests that mixing is occurring between groundwater that is associated with modern
recharge processes in the alluvium and groundwater that, as indicated by the $^{14}$C content, is
presumably much older. This older groundwater may be derived from artesian inflow. Figure
6 shows $^3$H activities > 0.04 TU in samples with $^{14}$C content of almost 0 pmc, suggesting that
groundwater with a very low $^{14}$C content is mixing with groundwater with a high $^3$H activity.
Even though there is evidence of $^{14}$C dilution in localised areas, we also observe mixing
between groundwaters of widely different $^{14}$C and $^{3}$H values in the gradient of the samples in
Figure 6 (emphasised with a dotted blue line). This gradient would be steeper if there were
mixing between groundwaters closer in residence times (Cartwright et al. 2013).

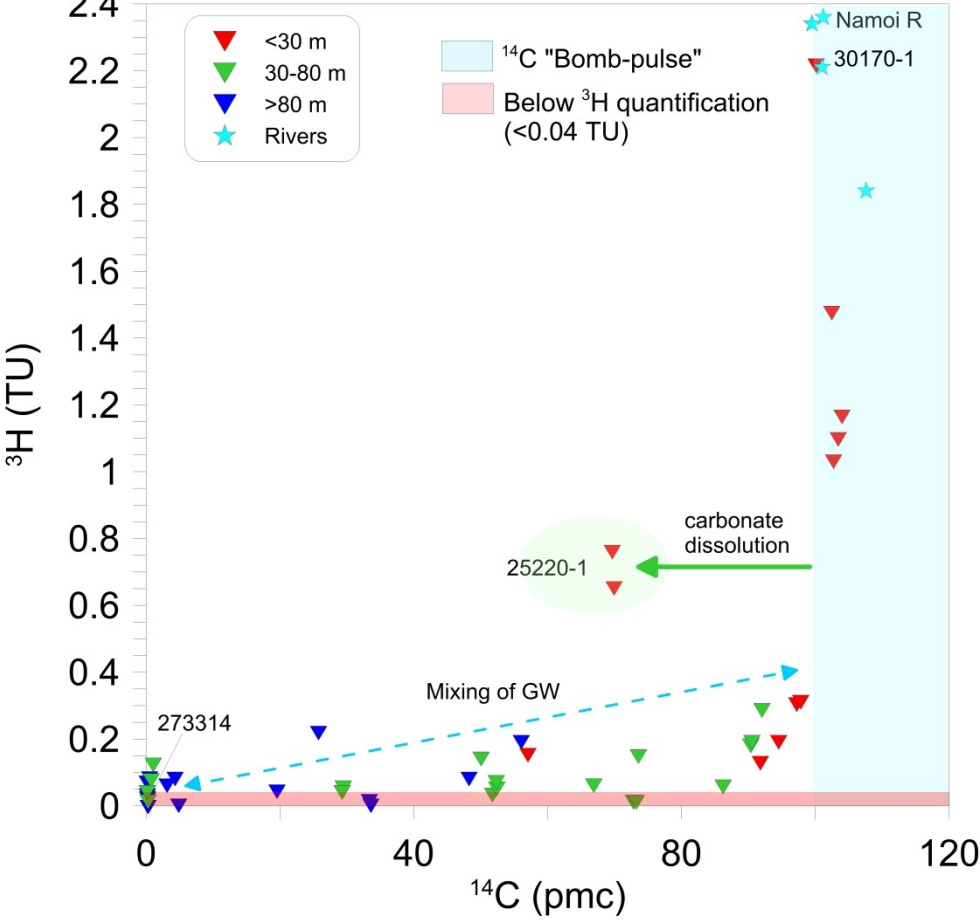


**Figure 6.** $^{3}$H (TU) vs $^{14}$C (pmc). This shows the mixing between groundwater with quantifiable $^{3}$H
activity (as indicated by the red band) and groundwater with very low $^{14}$C content (as indicated by the
dotted blue line).

**5.2    Extent of interaction between the GAB and the LNA**
The $^{3}$H and $^{14}$C values show that there is mixing between groundwater of varying residence
times, however they provide little constraint on the groundwaters with a $^{14}$C content of close
to 0 pmc (ie > 30 ka). This is where chlorine-36 dating can be a useful tracer because it can
be used to identify the presence of groundwaters that are much older than the range provided
by $^{14}$C.

A plot of $^{36}$Cl/Cl vs $^{14}$C (pmc) (Figure 7) shows a distinct mixing trend between

groundwater with high and very low $^{14}$C content. The 2 deep outlying samples (30345-2 and
273314; shaded yellow ellipse in Figure 7) display different geochemical characteristics from
the other samples, possibly because of their proximity to the Napperby Formation (Figure 2).
Figure 7 shows the $^{36}$Cl/Cl value range of GAB recharge, highlighting the alluvial samples
with values lower than this GAB recharge value. Calculations suggest that these particular
groundwater samples are potentially hundreds of thousands of years old, which is consistent
with groundwater from the GAB. This implies that these alluvial groundwaters are influenced
by artesian inflow of very old groundwater. This is evident in the natural neighbour
interpolation in Figure 2.

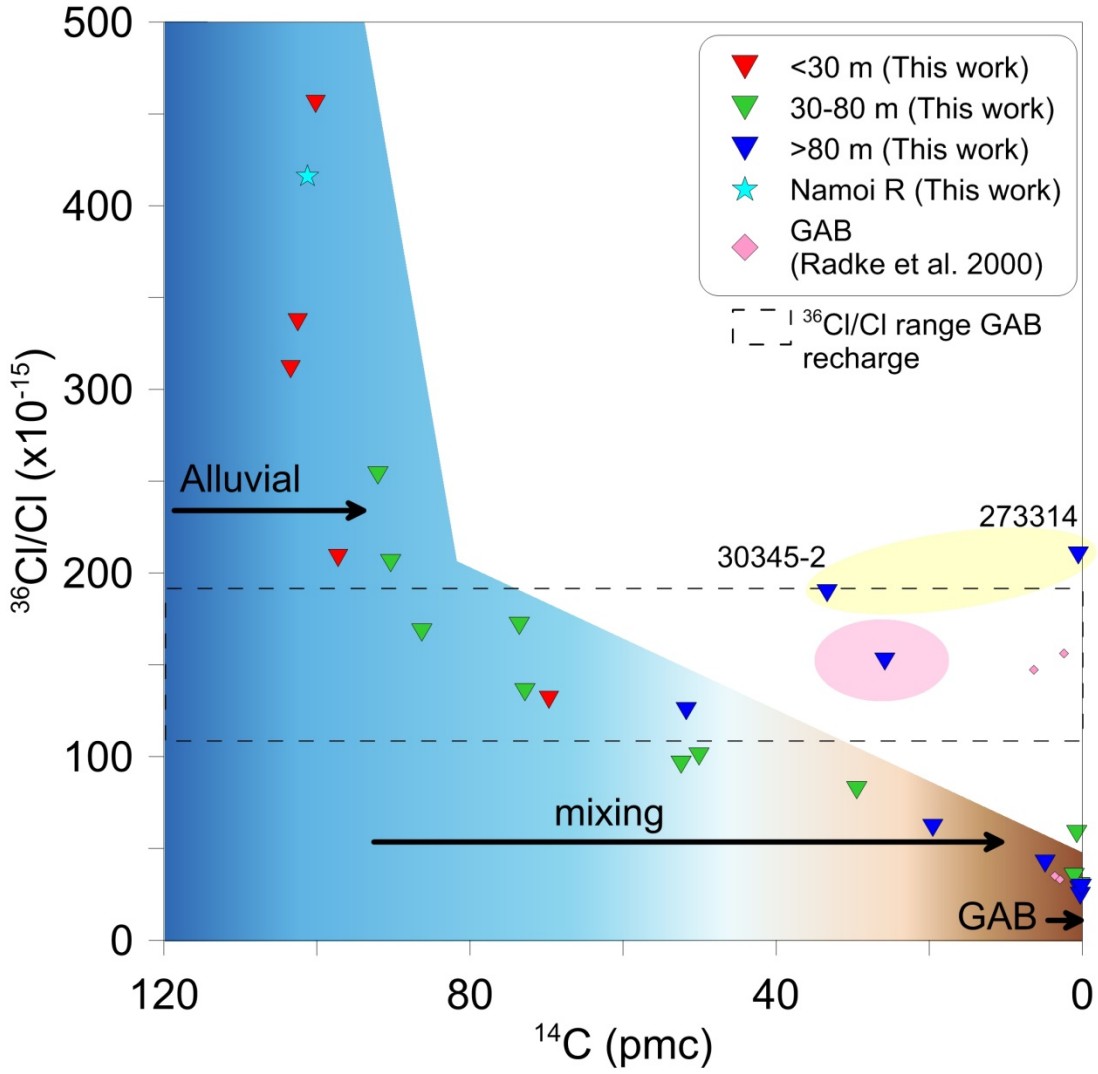

**Figure 7.** $^{36}$Cl/Cl (x10$^{-15}$) vs $^{14}$C (pmc). The colour gradient represents the mixing between the two major sources: surface water recharge (blue = modern) and the GAB (brown = old). The shaded yellow ellipse encompasses the two outliers where the geochemistry is being influenced by proximity to the Napperby Formation. The shaded pink ellipse is sample 25327-3 located in the irrigation area.

The apparent degree of $^{36}$Cl decay observed in the alluvial groundwater samples is too large to be explained simply by radioactive decay as indicated by the measurable $^{14}$C content in the same samples (Phillips 2000). This means that the time needed for the $^{36}$Cl to decay as much as observed would be well outside the range of $^{14}$C dating (> 30 ka) and therefore all groundwater samples would be expected to have a $^{14}$C content of 0 pmc, which is not

observed. Furthermore, the decrease in $^{36}$Cl is unlikely to result from dilution by $^{36}$Cl-
depleted sources such as evaporites, as the Cl$^-$ concentrations are similar in most samples
(Figure 8a and b). Therefore, mixing between groundwaters of different residence times is the
most likely explanation for the observed $^{36}$Cl signatures.

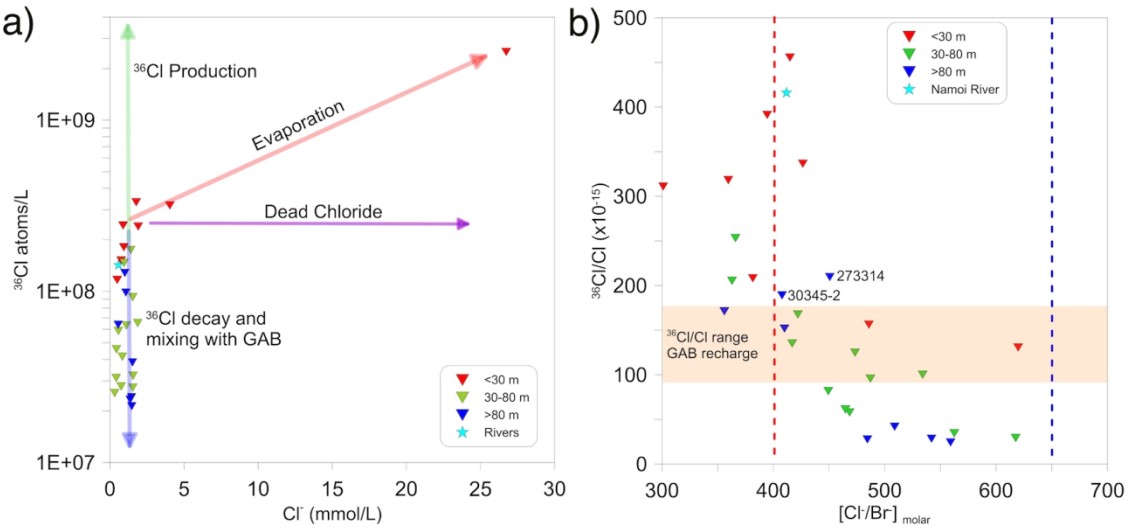


**Figure 8.** a) $^{36}$Cl vs Cl$^-$ concentration. The $^{36}$Cl production arrow represents potential in situ $^{36}$Cl
production from the high U and Th content in the host rocks; b) $^{36}$Cl/Cl ratio (x10$^{-15}$) vs Cl$^-$/Br$^-$. The
dotted blue line represents the Cl$^-$/Br$^-$ ratio in seawater and the dotted red line represents the expected
Cl$^-$/Br$^-$ ratio for rainfall at Narrabri based on distance from the coast (Short et al. 2017).

Our groundwater samples from the deep alluvium display lower $^{36}$Cl/Cl ratios (down to
24 (x10$^{-15}$)) than those measured in the GAB recharge zone. This indicates that there is very
old groundwater in the deeper LNA (conceivably older than that of the GAB recharge zone),
and that the mixing observed in our geochemical data could be taking place between
groundwater with a residence time of less than 70 years (assumed using $^3$H) and groundwater
with low $^{36}$Cl activities, consistent with GAB groundwater that is potentially hundreds of
thousands of years old (Radke et al. 2000).
To quantify the extent of interaction between the two groundwater sources, we use the
concentration of the conservative chloride ion to determine an approximate percentage of GAB
to alluvial groundwater at each sample location. In general, Cl concentrations in surface water
and shallow groundwater in the study area are low (< 30 mg/L), while samples recovered from
the Pilliga Sandstone (GAB) have higher concentrations (~ 60 mg/L). To estimate the local
surface infiltration end-member, a shallow groundwater sample with a high $^3$H activity (sample
30170-1; 2.21 TU) was used. The average of all available GAB data was used for GAB inputs.
These end-members are mixed in varying proportions to obtain the Cl$^-$ concentration that we
observe in all our groundwater samples (via inverse modelling calculations). If the Cl$^-$
concentration in the sample was lower than that in the representative local surface infiltration
sample, a 100% LNA contribution is assumed. The representative sample used as the local
surface infiltration end-member has been subject to some evaporation and therefore does not
have the lowest Cl$^-$ concentration in the alluvium. If the sample with the lowest Cl$^-$
concentration was used as the surface water end-member, we would require a higher percentage
of GAB contribution across the study area. Thus, the use of the evaporated sample as our end-
member represents a conservative approach when considering the mixing components from
both the LNA and the GAB.

The Cl mixing results provide an approximate mixing threshold with shallower samples

generally containing a higher proportion of alluvial groundwater, which diminishes with
depth. These mixing proportions show that some deeper samples in the LNA contain up to
70% GAB groundwater. Figure 9 presents approximate contours for artesian discharge
proportions into the LNA based on the Cl mixing approach. The dotted lines indicate areas
where there is just one sample to inform the interpretation, whereas the solid lines connect
multiple samples that all displayed similar contributions from the GAB.

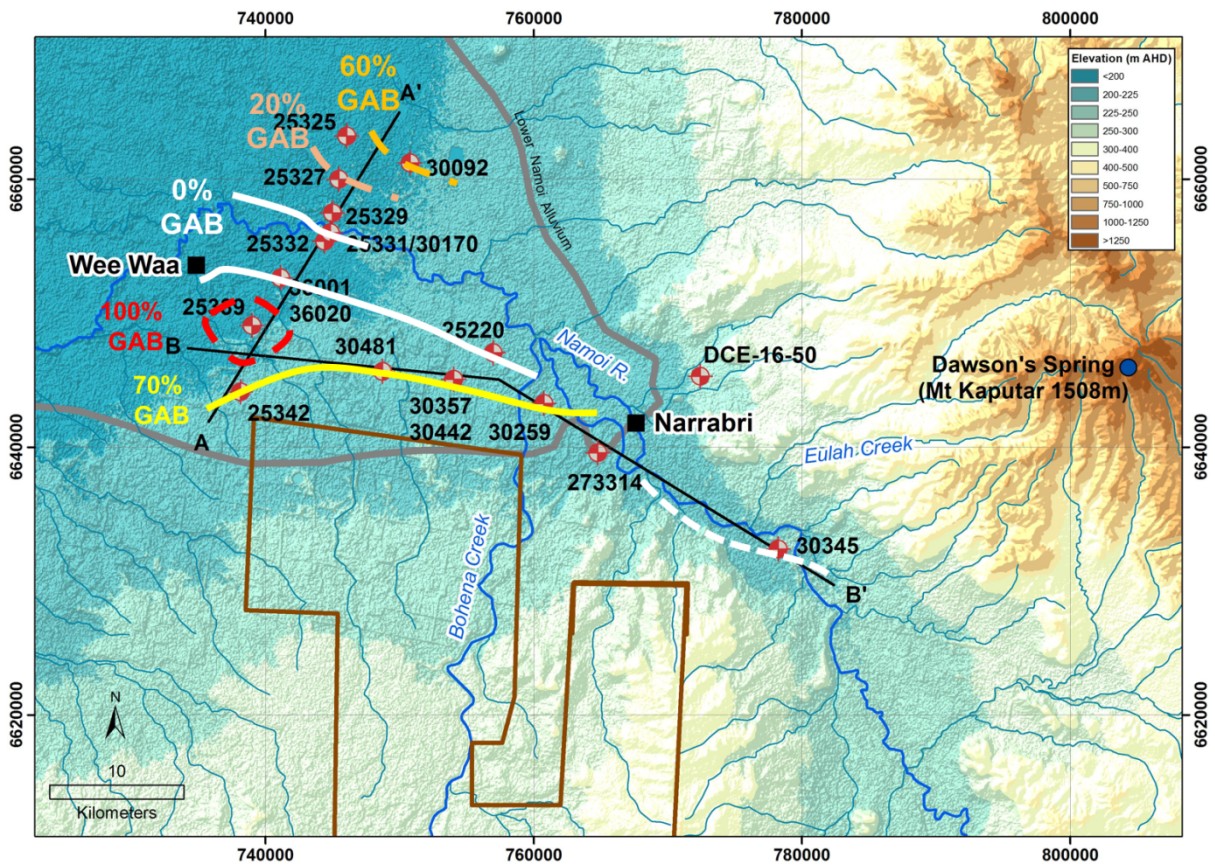

**Figure 9.** Approximate percentages of GAB contribution to the LNA, calculated from multiple geochemical tracers and major ion data.

Artesian input can be inferred from nested piezometers at locations 30481 and 30259 (Figure 1). At these locations, the monitoring bore slotted in the lower portion of the LNA has a head higher than the monitoring bore slotted in the shallow portion of the LNA, indicative of upward flow. At all other locations artesian contributions cannot be discerned from head data. Comparing Figure 9 to Figure 1 we show that groundwater geochemistry can provide a more accurate evaluation of GAB contribution to the LNA. This is because the geochemical data can elucidate groundwater mixing processes and provide longer term insights compared to the hydraulic head data. Multiple geochemical tracers reveal that boreholes in the north and west of the study area may be experiencing much more GAB inflow than has been inferred in catchment water balance models (Merrick 2000; Kelly et al.

2007; CSIRO 2007). This is most evident at sample 25342. It is not immediately apparent
from the vertical heads in the hydrograph set at sample 25342 that there is any GAB inflow,
yet based on the geochemical tracers this location is 100 % GAB groundwater. The water
balance model described in Merrick (2000) has GAB groundwater contributing 22% of all
inflow into the LNA between Narrabri and Wee Waa (Figure 1). From the geochemistry
alone it is not possible to make an estimate that can be directly compared to that artesian
discharge estimate. However, it is apparent from the mixing results shown in Figure 9 that a
large portion of the study area has an artesian input to the LNA that is likely to be greater
than 22%. The above observations highlight why geochemical insights should ideally be used
as one of the constraining data sets when doing water balance models in regions where there
is both artesian discharge and surface water recharge to the alluvial aquifer.

**5.3     Temporal changes in the interaction between the LNA and the GAB**
The multiple geochemical tracers we have used show substantial artesian discharge to the
LNA, which is larger than that currently considered in groundwater models of the region
(Merrick 2000; Kelly et al. 2007; CSIRO 2007). However, it is difficult to constrain how the
extent of artesian discharge has changed over time and how it may continue to change. Time
series sampling can constrain how this GAB discharge will change and is important for
understanding future artesian contributions to the LNA. Past $^{14}$C (pmc) data collected from
the same bores in 1978 (Calf), 2003 (McLean), 2010 (ANSTO data) and 2016 (this study)
enable us  to observe how the $^{14}$C content in the groundwater has changed over time. The
historical $^{14}$C data, coupled with data from this study, has the potential to be used as a
preliminary indicator of changes in the relative contributions of high $^{14}$C contents from recent
surface recharge (~ 100 pmc) versus low $^{14}$C contents of the GAB discharge to the LNA. The
dataset contains 14 bores from 5 nested sites and is the most comprehensive long-term time-
series database for the study area, if not Australia, despite not being complete for all years.

Most of the samples displayed relatively consistent $^{14}$C values across the years where

data were available. However, we observed large changes in $^{14}$C content in 5 monitoring
bores; 4 showed an increase and 1 showed a decrease (bold text in Table 1). This suggests
that the varying contributions of older and younger groundwater has changed over time,
which could be a preliminary indicator of increased surface recharge to various sites, or
increased artesian discharge to others. Therefore, measuring the $^{14}$C in the groundwater at any
future time and assessing how this has changed using past data is useful as a preliminary
indicator for the current state of the system. However, consistent data collection and
incorporation of other factors that may affect groundwater mixing (such as rate of
groundwater extraction and amount of surface infiltration) are necessary to make inferences
about temporal changes in the interaction between the LNA and the GAB.

**Table 1.** Changes in $^{14}$C content (pmc) in select boreholes in the study area between 1978-2016 (see
Figures 1 and 8 for the locations of the bores). The 5 bores in bold text highlight where we observe
changes in the $^{14}$C content from 1978 to this study. Where available, the time of sampling is included.

ND = no data.

| Bore | Depth interval (m bgs) | Calf (1978) | McLean (2003) | ANSTO data (summer 2010) | This study (summer 2016) | This study (winter 2016) |
|---|---|---|---|---|---|---|
| **25220/1** | **24.4-30.5** | **28.15** | **ND** | **ND** | **69.66** | **69.94** |
| 25220/3 | 97.5-109.7 | 0.99 | ND | 0.13 | 0.17 | 0.22 |
| 25325/2 | 36.9-38.4 | 83.63 | ND | 85.77 | 86.25 | ND |
| **25325/6** | **67.1-70.1** | **65.31** | **ND** | **66.57** | **90.37** | **ND** |
| 25332/1 | 17.7-21 | 103.61 | ND | ND | 102.48 | ND |
| 25332/2 | 38.1-41.1 | 99.19 | ND | 104.78 | ND | ND |
| 25332/3 | 50.9-55.5 | 94.70 | ND | ND | ND | ND |
| **25332/4** | **66.8-69.8** | **49.33** | **ND** | **84.12** | **73.57** | **ND** |
| 25327/1 | 18.9-21.9 | 123.36 | 101.3 (s) | ND | 103.43 | 102.74 |
| 25327/2 | 57.9-60.9 | 84.16 | 93.78 (s) | ND | 92.05 | 90.56 |
| **25327/3** | **80.8-83.8** | **8.48** | **8.63 (s)** | **ND** | **25.79** | **56.08** |
| 30092/1 | 17.7-20.7 | ND | 90.51 (w) | ND | ND | ND |
| **30092/2** | **48.2-49.4** | **ND** | **80.06 (w)** | **72.31** | **ND** | **66.92** |
| 30092/4 | 108.2-110 | ND | 0.19 (w) | 0.24 | 0.3 | 0.21 |


## 6    Conclusion

We have used multiple geochemical tracers to show that artesian discharge to a shallow
alluvial aquifer is higher than previously derived from water balance models in the literature
(Merrick 2000; CSIRO 2007; Kelly et al. 2007). We have also provided a percentage
estimate of GAB groundwater in each sample collected in the LNA using the concentration
of Cl in the groundwater, showing that in some locations the 'alluvial' sample is comprised
of up to 70% GAB groundwater. Our findings are important when considering the global
importance of groundwater and the sustainable use of connected alluvial and artesian
systems, globally.
Isotopic tracers ($^{3}$H, $^{14}$C, and $^{36}$Cl) indicate that there is substantial mixing between two
groundwater end-members of very different residence times (< 70 years and very old
groundwater consistent with the GAB). This suggests interaction between modern surface
recharge through the shallow LNA and variable artesian inflow at depth, dependent on where
the sample is located in the system. We have also used past $^{14}$C data (1978, 2003, 2010),
along with data from this study to show that these data can be used as a preliminary indicator
of how the extent of interaction between the GAB and the LNA has changed over time. Yet,
how these trends change geographically throughout the system, and how they will behave in
the future are difficult to constrain without continuous monitoring.
In the interval of the Lower Namoi studied discharge from the GAB into the LNA was
previously considered to contribute approximately less than 22% of the input water to the
LNA (Merrick 2000; CSIRO 2007; Kelly et al. 2007). However, the geochemical data
reported above clearly indicate that GAB discharge is occurring in locations where inflow is
not apparent from the nested hydrograph data. This highlights the need to apply multiple
groundwater investigation techniques (including flow modelling, hydrograph analysis,
geophysics, and geochemistry) when inferring artesian discharge to an alluvial aquifer. This
research has demonstrated that a multi-tracer geochemical approach is required to better
determine artesian contributions to the alluvial aquifer and must be considered in
constraining future models of the study system and elsewhere.

**Acknowledgements**
This research was funded by the Cotton Research and Development Corporation (CRDC).
Charlotte Iverach was supported by scholarships from the Australian Government, ANSTO
and CRDC. ANSTO support and analytical staff are thanked for their continuous efforts
(Chris Dimovski, Henri Wong, Robert Chisari, Vladimir Levchenko, Krista Simon, Alan
Williams, Simon Varley). The authors also thank Dr. Lisa Williams for editing and
proofreading the manuscript. In addition, many thanks to the associate editor (Markus
Hrachowitz) and the three reviewers, who provided constructive feedback and raised the
overall quality of the manuscript.

**Author contributions**
Experimental conceptualisation and design was carried out by D.I.C & B.F.J.K. Fieldwork
was conducted by C.P.I., D.I.C., S.I.H. & B.F.J.K. Additional data was contributed by
K.T.M. Geochemical analyses were conducted by C.P.I., D.I.C. & K.M.W. The manuscript
was written by C.P.I with input from all authors.

**Competing Interests**
The authors declare that they have no conflict of interest.

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

**List of Figures**
**Figure 1.** Map of the study area and sample locations, along with the location of the study
area in Australia. Accompanying hydrographs show the groundwater level response in
different piezometers throughout the study area (groundwater level data sourced from BOM
2017). The different colours in the hydrographs represent the different monitoring bores in
the nested set. The bottom of the slotted interval for each bore is shown in the key. The x-axis
in each hydrograph is the year (1970-2010) and the y-axis is depth (between 0 and 40 m
below ground surface (bgs)). The two locations with red text highlight areas where the
hydrograph heads show clear GAB contribution, with the deeper piezometer showing a
higher head than the shallow one. The remaining locations show no apparent GAB
contribution to the LNA based on the hydrograph data.
**Figure 2.** A geological map of the study area and two cross sections through the study area,
showing the location and depth of the samples in the alluvium and their proximity to
formations of the GAB. Contacts obtained from gas wells Nyora-1, Culgoora-1 and
Turrawan-2, coinciding with our cross sections, are added. Their locations are displayed on
the map. The general direction of groundwater flow is from SE-NW, aligning with the B-B'
line on the map. The chlorine-36 data interpolated using the 'natural neighbours' algorithm is
shown in each cross section.
**Figure 3.** a) $Na^+$ vs $HCO_3^-$ showing the mixing trend that the alluvial samples form between
the Namoi River and samples from the GAB (Radke et al. 2000; McLean 2003). The orange
calcite saturation line indicates samples that are more enriched due to separate
evapotranspiration and calcite precipitation. The shaded blue ellipse represents all river
chemistry data available for the Namoi River and tributaries (this work (n=4), McLean 2003
(n=4), Mawhinney 2011 (n=79)); b) $Na^+$ vs $F^-$ and c) $Cl^-/Br^-$ vs $Cl^-$, highlighting the mixing
trend between the surface recharge and the GAB that we observe in other geochemical
indicators. The red dotted line represents the $Cl^-/Br^-$ ratio for rainfall and the blue dotted line
is the seawater ratio.
**Figure 4.** Water stable isotopes in the LNA, showing the two separate mechanisms of
recharge; surface water recharge plotting along an evaporation trend line and potential inflow
from the GAB clustered with regional samples from the GAB (McLean 2003).
**Figure 5.** Plot of depth vs $^3$H, highlighting the $^3$H activity throughout the vertical
groundwater profile. Samples that fall within the pink zone on the left are below the
quantification limit (< 0.04 TU). These data are not included in our interpretation of how $^3$H
changes with depth. They are presented to convey the relative proportions of interpretable
versus non-interpretable data.
**Figure 6.** $^3$H (TU) vs $^{14}$C (pmc). This shows the mixing between groundwater with
quantifiable $^3$H activity (as indicated by the red band) and groundwater with very low $^{14}$C
content (as indicated by the dotted blue line).
**Figure 7.** $^{36}$Cl/Cl (x10$^{-15}$) vs $^{14}$C (pmc). The colour gradient represents the mixing between
the two major sources: surface water recharge (blue = modern) and the GAB (brown = old).
The shaded yellow ellipse encompasses the two outliers where the geochemistry is being
influenced by proximity to the Napperby Formation. The shaded pink ellipse is sample
25327-3 located in the irrigation area.
**Figure 8.** a) $^{36}$Cl vs Cl$^-$ concentration. The $^{36}$Cl production arrow represents potential in situ
$^{36}$Cl production from the high U and Th content in the host rocks; b) $^{36}$Cl/Cl ratio (x10$^{-15}$) vs
Cl$^-$/Br$^-$. The dotted blue line represents the Cl$^-$/Br$^-$ ratio in seawater and the dotted red line
represents the expected Cl$^-$/Br$^-$ ratio for rainfall at Narrabri based on distance from the coast
(Short et al. 2017).
**Figure 9.** Approximate percentages of GAB contribution to the LNA, calculated from
multiple geochemical tracers and major ion data.