# Peer review of "A multi-tracer approach to constraining artesian groundwater discharge into an alluvial aquifer Authors: Charlotte P. Iverach1,2,3, Dioni I. Cendón1,2,3, Karina T. Meredith3, Klaus M. Wilcken3, Stuart I. Hankin3, Martin S. Andersen1,4, Bryce F.J. Kelly1,2,\* 1Connected Waters Initiative Research Centre, UNSW Sydney, NSW, 2052, Australia 2School of Biological, Earth and Environmental Sciences, UNSW"

_Hydrology and Earth System Sciences, 2017_

## Referee Comment (RC1) · Anonymous Referee #1 · 14 Jul 2017

This is potentially an interesting study with important results and HESS would be a suitable place for publication. I started making detailed comments on the earlier sections (and have included them below as they may be useful) but once I got to Section 4, which represents the bulk of the scientific discussion, I came to the conclusion that the paper needed major rewriting and reorganization before it was suitable for publication. In its present form it is difficult to comprehend and I do not think it would be impactful.

I have outlined some of the problems below together with suggestions.

The results and discussion section is poorly set out and not convincingly written. For example, take the text near the start of section 4.1 (line 343):

"There is an excess of both Na+ and HCO3- in the groundwater of the LNA (Supplementary Table 2), compared to ion ratios expected from local rainfall sources and other shallow groundwater alluvial systems in eastern Australia (Martinez et al. 2017). Their abundance defines the ubiquitous presence of Na-HCO3-type groundwater we observe throughout the study area. The Na-HCO3 ratio in GAB groundwater is generally 1:1 (ppm) (Radke et al. 2000; McLean 2003), which is reinforced by the position of the regional GAB samples in Figure 3a. The Namoi River and other regional streams have lower Na+ and HCO3- concentrations and a lower Na+/HCO3- ratio than both the historic GAB data and the deeper alluvial data collected in this study."

This paragraph makes conclusions without explaining their basis and does not adequately describe the data (it just points the reader to tables and figures where the data are summarized / plotted). There are also several concepts mixed together (the ion excesses, the comparison of water from different reservoirs).

The next part of this discussion (lines 351 to 362) alsomixes observations with conclusions and deals with a variety of processes (mixing, evapotranspiration, and calcite saturation). There are again few details; what does "towards calcite saturation" mean (what are the saturation indices, where do you discuss them)? Why is the calcrete important (is it found at groundwater discharge points in the watershed, for example?). I agree that the halogen chemistry in Fig. 4 is consistent with evaporation but you need to explain why that is so in the text (perhaps more to the point is this important for understanding recharge and mixing?).

Most of the other sections / paragraphs in the results and discussion are similarly hard to follow. For example the discussion of Tritium (lines 416 to 431) includes discussion on historic Tritium concentrations with the distribution of Tritium from this study. The latter is never illustrated (the data are in a supplemental table but the spatial variation is important so should go on a map or section). The Tritium activities are not specified so sentences such as "However, despite decreasing activities, 3H remains relatively prevalent in the deeper part of the system." are very non-specific. The conclusion

reached "This indicates the extent of recharge from episodic flooding and shows that surface recharge reaches the deeper LNA (down to  $\sim$ 80 m bgs) relatively quickly (< 70 years)." then becomes impossible to assess.

There has been a lot of work on determining residence times and mixing from concentrations of radioactive isotopes (including by the coauthors of this paper). Much more could be done to firm up the conclusions, for example some of the samples on Fig. 5 seem to not have undergone extensive mixing and probably could be used to determine residence times.

In section 4.3 it is not clear whether the Chlorine-36 data are being interpreted in terms of ages or mixing. Elsewhere, you have stressed mixing but here you calculate ages. This is despite Chlorine-36 being notoriously difficult to use for anything other than broad indications of residence times due to the input function varying (in unknown ways) over time due to climate variations. The Chlorine-36 ages are presented without much skepticism or discussion.

Calculations through Section 4 are poorly presented. For example, the discussion of mixing (lines 509-530) uses a single composition for the end-members and these is no sensitivity analysis. The results are presented without much discussion of uncertainties etc.

Section 4 is not helped by its structure. It mixes introductory material (e.g., lines 453-455), conclusions, and description. It is also not very tightly written and uses mainly qualitative descriptors rather than specific values. It is possible to mix results and discussion, but it needs skill otherwise the text becomes meandering and there is commonly not the rigor in explaining the salient features of the data before they are interpreted. The conclusions of this paper are plausible, but the way that it is written does not do them justice.

Some suggestions for revamping the paper are:

1) Separate the results and discussion and make sure that you adequately describe the important pieces data (don't just say which diagram or figure it is contained in).

2) Concentrate on what is important. The aims of this study is to understand recharge and mixing, in which case some of the details of the water chemistry seem superfluous. For example, it is important to determine whether the GAB waters and local recharge have different compositions but some of the details of the processes could be omitted. The Chlorine-36 is more valuable as an indicator of mixing rather than residence time, but recharge rates estimated from Tritium are important.

3) In a similar vein, the paper would be improved by more hydrologic information at the expense of some of the detailed geologic information. Is the interpretation of mixing consistent with the hydraulic heads? What are the groundwater flowpaths?

4) Include enough justification of calculations to make them convincing and some sensitivity analysis or discussion of uncertainties.

I am guessing that the senior author is a graduate student. The coauthors, however, are not and should have picked up on the more obvious problems with the way that this study was framed and presented.

Specific comments

Introduction

The introduction provides a general outline of the science and the reasons for carrying out the study. The first paragraph is not very clearly expressed. For example ć Why specify "modern infiltration" – recharge implies modern processes ć "Spatial and temporal data resolution and heterogeneity in hydrogeological properties result in considerable uncertainty when allocating recharge to each source and mapping pathways of flow" is very unclear. ć What is a "dynamic groundwater gradient"?

The introductions to papers are important as they frame the study and hopefully persuade the reader to continue reading, so it is worth making them as clear as possible. Line 86. What do you mean by "modern/submodern"?

Line 89. It would be useful to explain briefly how the various isotopes help understand mixing as it might not be clear to all readers.

Line 96. It would be clearer if you split this material off as your objectives get lost at the end of the discussion of the techniques (perhaps put a subheading in for emphasis).

Lines 98-99. This is stating a conclusion, which you should leave until later.

Study Area

This section provides a comprehensive description of the study area. Some specific comments

Lines 139-141 is difficult to follow for anyone not familiar with eastern Australian groundwater hydrology. Can you add a key map of the basins to Fig. 2?

Lines 141-162. The description of geological framework is difficult to relate to Fig. 2 as you do not specify the age of the various units. Provide a few more details in the text.

Section 2.1 is probably too detailed for the study. While understanding the geology of the area is important, I am not convinced that the geologic history needs to be gone through in this much detail (for example, is the rainfall variation between the Miocene and today important for this study). This section could be cut fairly substantially. What would be more useful, and which is not there, are firstly some hydraulic properties (K, porosity etc) and secondly some description of groundwater flow.

Lines 197-201. This repeats material in the introduction and could be removed.

Line 205. The numerous Merrick references seem to be to a series of non-publically available documents. The reference to the Kelly et al., summary would seem sufficient.

Lines 208-209. Not clear what you mean by "There are equivalent solutions for all water balance models and the solution presented is often constrained by several factors."

Section 2.2.1 does not add that much useful. You state that there are a range of models but provide few details. The discussion of the models seems to reside only in unavailable consultants' reports and then the point about not taking into account geochemistry is reiterated. This section could be shortened, especially as you do not make detailed comparisons with specific models later in the paper.

**Figures**

Fig. 1. Define "bgs". Rather than describing what is on the two axes, just label them in the graphs. Text on the map is too small (you could make the map larger and put the Australian map in an unused corner)

---

## Referee Comment (RC2) · Anonymous Referee #2 · 24 Jul 2017

General Comments: This manuscript has potential to be a very interesting article and could have a big impact on our current understanding of the interactions between alluvial aquifers, regional aquifers, and surface-water systems. The topic should have broad interest to other research on alluvial aquifers and broad applicability to global alluvial groundwater systems. The combined geochemical tracer and multiple isotopic tracer approach is a potentially robust way to sort out these interactions. However, the manuscript suffers from poor organization and lack of specificity in the methods and interpretations. This prevents a recommendation to accept the article as it is currently written. Instead, the recommendation is to accept after major revisions including the organization and technical detail of the article. In addition, it was very difficult to read

the manuscript due to awkward syntax and lack of focus (see Specific Comments).

The major findings/contributions of this manuscript hinge on the authors' interpretations of the isotopic and geochemical data. In general, these sections need much more clarification than is provided and in some cases, additional data mining may be ncessary (although I think the latter is the smaller of the issues). For example, a considerable emphasis is placed on 3H concentrations. However, almost all of the 3H concentrations are close to background or are 3H dead. This has implications for the mixing model and it has implications for the residence time estimates. It's not unusual to find 3H dead waters which have slightly elevated 36Cl/Cl ratios. 3H, even from the bomb-pulse, is decaying faster than 36Cl even in the presence of mixing with recent recharge and sometimes 36Cl does a better job of sorting out mixing than 3H (especially for waters that are 100 years or older). However, the real strength that the authors have is the ability to sort out young and old fractions (relative terminology) using 14C pmc. In the absence of carbonate exchange and mixing in the aquifer (another topic which needs additional clarification) the 14C pmc value is very useful in sorting out these endmembers as the authors attempt to do in Figures 5 and 6. In any case, the authors need to provide additional clarification on how they sorted out the mixing processes that affect these tracers.

Regarding data mining, this manuscript and the authors' interpretations would be strengthened by providing the 3H, 13C, and 36Cl/Cl of modern precipitation. Are these data available? If so, cite them. This would enhance the mixing model which is used to determine the fractions of GAB water in the LNA. This calculation also needs additional clarification and the authors should present the equation and discuss sources/estimates of uncertainty in this calculation.

Specific Comments: Line 56: This paragraph lacks focus. The authors bring in agriculture and it's not clear how this is connected to the bigger issue.

Line 64: What is meant by the international export market?

Lines 65-78: What is the focus here? Aquifers in general or alluvial aquifers?

Line 81:It's not just the half-life that is important, the tracer systematics including mixing and processes that affect their interpretations are needed.

Lines 96-105: The manuscript needs a focused statement about the current knowledge gaps (this should be developed in the Introduction more succinctly) and what is new and novel about this research in addressing those gaps. The manuscript would benefit from a clear hypothesis statement or statement of science questions. This, in my opinion, would help focus the entire manuscript and the authors should return to this statement in the opening paragraph of the Conclusions.

Lines 208-213: This sentence is difficult to understand, yet this paragraph is critical in identifying the knowledge gaps.

Line 237: This is a good place to re-state or reiterate what is missing by identifying the knowledge gaps and how your research addresses those gaps. Lines 292-293: Please clarify the statement on NH4 concentrations.

Line 336: Does it make more sense to separate the Results and Discussion. This may help streamline and better organize the manuscript.

Lines 343-362: This needs to be better organized, perhaps start with description of how GAB works hydrologically, then describe the ratios and their implications for the LNA. Suggest breaking this paragraph into 2 new, concise paragraphs.

Lines 375-376: Be assertive here. This does suggest...rather than may suggest.

Lines 376-378: Needs clarification.

Lines 380-381: Please provide sources of F- and how that relates to the point you are making.

Lines 416-431: Please clarify. Most 3H values are close to background or are dead with respect to 3H. This complicates your interpretation, but you seem to pull it back

in focus with the figure. Suggest picking specific sites and describe what the data is telling you.

Lines 432-434: Prevalence of 3H?? Again, the 3H values are almost all very low or 3H dead. Consider rewording and clarifying this statement. Also, are there recent 3H values for precipitation?

Line 434: 3H and 14C and not entirely consistent are they? This needs clarification. Why are they inconsistent?

Lines 432-445: Please clarify and provide a more concise discussion on mixing effects.

Line 458: What about 36Cl/Cl of modern precipitation? High 36Cl/Cl can be indicative of mixing of bomb-pulse with recent (low 36Cl/Cl) recharge. But, low 36Cl/Cl can also imply very old groundwater (your case). Can you clarify this uncertainty with modern precip?

Line 487 (Figure 7): Please cite Phillips (2000) Chapter 10.

Lines 510-523: Please provide the equation used to calculate the mixing proportions. Can you also provide estimates of uncertainty in these calculations? Are there other solutes or solute ratios (Cl/Br for example) which may be more suitable for these calculations? It's not clear how this was estimated.

Line 535: Can you provide plots showing the spatial map of appropriate chemical concentrations?
* * *

---

## Referee Comment (RC3) · Anonymous Referee #3 · 2 Aug 2017

The presented study is well designed and informative for regions, where different water bodies seem to exist and mix in ratios, which are unknown yet. Thematically the paper fits to HESS, although I see some points of weakness, mainly related to formulation (or omitting) of hard facts. I guess, with considerable revision, that manuscript has the potential to be of interest for a wide audience. In general, the manuscript should be shortened and particularly the geological part must be clarified for readers outside Australia. In the following, I give some specific remarks to points, where I see difficulties:

Hydrographs in Figure 1 are not very informative, despite the information, that gwtables are fluctuating. the legends of hydrographs are not explained and it becomes not obvious, why red-texted hydrographs are representative for GAB contribution. Instead of showing relative depths of screen bottoms (bgs), it would be more distinctive, when depths would be given relative to msl., to explain the absolute depth.

Hydrogeological setting The entire paragraph is very hard to understand, since local formation names are abundant and the hydrogeological context is not clear. Why are all these details neccessary for the reader of the manuscript (e.g. lines 191-193)? Paleogeorgraphic features are very difficult to understand. It would be of more importance to reduce the (doubtless interesting) geological context and focus on the formations, which are hydraulically relevant. Probably a stratigraphic table would help a lot, showing thickness, lithological composition and phreatic/confined conditions in each of the relevant formations.

221 Water balance modeling for recharge That paragraph explains a series of MODFLOW attempts to define various sources for recharge. I believe, the paragraph is to long, since the basic and neccessary information are the outcoming numbers (ratios) for the different proposed sources. The authors use a unit (ML/a) which is unknown to me (Megalitres/year?)

3.2 Geochemical analysis Line 302: what is the reason to use pmc and pMC?

4 Results line 358: whic 2 processes are meant? ET leads to enrichment of all elements, leading eventually to Cc-saturation. Na/HCO3 increases only, when calcite precipitates.

line 360: I suggest to be careful in interpreting Cl/Br ratio changes in these context. Cl/Br ratio will change only, when degree of evaporation results in supersaturation of the water in respect to halite, otherwise there is no change observable. Since Cc-precipitation is discussed, it might be worthwhile to compare Ca/Mg ratios and (Ca+Mg)/HCO3 ratios? Cl/Br ratio might change due to geological reasons...

line 390: delete charges. What means "closer"? compared to what?

line 399-401: that sentence is not helpful, since the reader does not know which parameters you refer to. From Figs 3 and 4 it is not given, that 273314 resembles river water, it is obviously just fresh water. line 401 ff: from that moment it becomes highly difficult to follow: you refer to the only sample from the Jurassic Fmt. Why is it strange to have fresh water in there? The base of the well is just above a Napperby fmt. Which indicators suggest recharge through a formation, which is even below Napperby? And which river is referred to? Why should Pilliga Sandstone contribute? The explanation lacks from facts, which give an overview about the hydraulic concept, which obviously led to the formulations. Latest here a regional W-E geological cross-section, showing Fmts. of GAB and their regional confined and phreatic conditions (piezometer heights) is urgently needed to understand the hydrogeological context of the region. In addition, it would also help, to (i) show Fmt. and (ii) add water table heights of the different aquifers in the cross-sections of Fig. 2. Situation becomes harder due to the jumping between formation names.

line 407 ff: again, why do the authors claim for contact between that river and deeper Triassic Fmt.? According to Fig. 2: Napperby is the uppermost Triassic. Where is that river situated and why is the river the only option of fresh-water supply? Are these ideas consistent with hydraulic?

4.2. Mixing

line 412f. : To be very critically: I don't see clear indications for that statement from Figs. 3 and 4. Major elements in samples >80 m (blue) spread over the entire range and only a few blue samples fall in the same region as GAB analyses from Radke et al. (2000). Is there a geographic link?

4.3 Extend of interaction

line 517f. : Why is not a sample chosen, which was not evaporated at all or even better,

a recent rainfall sample, giving the precise input signal for Cl and 3H?

lines 521-523: I do not understand the reason of that thought: "...to consider overall transport of Cl from shallow groundwater."

line 532/fig. 8: Actually these percentages are calculated on Cl-mixing approach only. Within the description, "multiple geochemical tracers and major ion data" are mentioned. Which exactly were used and how does the respective results fit to the described Cl-mixing?

According to that figure, it strikes, that heterogeneity of GAB contribution might be related to structural features or any other elements that provide preferential flow? Are there any tectonic lineaments or other indications, which could be responsible for the different contributions from the GAB?

5. Conclusions

lines 619-621: That sentence is very vegetarian, it gives no information at all. Please prevent to use such phrases, instead of describing which reason will result in which effect.

---

## Author Comment (AC2) · 13 Sep 2017

The comment was uploaded in the form of a supplement:
https://www.hydrol-earth-syst-sci-discuss.net/hess-2017-327/hess-2017-327-AC2-supplement.pdf

---

## Author Response (AR1)

**This is potentially an interesting study with important results and HESS would be a suitable place for publication. I started making detailed comments on the earlier sections (and have included them below as they may be useful) but once I got to Section 4, which represents the bulk of the scientific discussion, I came to the conclusion that the paper needed major rewriting and reorganization before it was suitable for publication. In its present form it is difficult to comprehend and I do not think it would be impactful. I have outlined some of the problems below together with suggestions.**

We thank the reviewer for taking the time to review our manuscript closely. We have accepted the majority of suggestions detailed below, in particular the suggestion to reorganise the results-discussion sections. Throughout this response reviewer comments are bold whereas our responses are normal font.

**The results and discussion section is poorly set out and not convincingly written. For example, take the text near the start of section 4.1 (line 343):**

**"There is an excess of both Na+ and HCO3- in the groundwater of the LNA (Supplementary Table 2), compared to ion ratios expected from local rainfall sources and other shallow groundwater alluvial systems in eastern Australia (Martinez et al. 2017). Their abundance defines the ubiquitous presence of Na-HCO3-type groundwater we observe throughout the study area. The Na-HCO3 ratio in GAB groundwater is generally 1:1 (ppm) (Radke et al. 2000; McLean 2003), which is reinforced by the position of the regional GAB samples in Figure 3a. The Namoi River and other regional streams have lower Na+ and HCO3- concentrations and a lower Na+/HCO3- ratio than both the historic GAB data and the deeper alluvial data collected in this study."**

**This paragraph makes conclusions without explaining their basis and does not adequately describe the data (it just points the reader to tables and figures where the data are summarized / plotted). There are also several concepts mixed together (the ion excesses, the comparison of water from different reservoirs).**

The results and discussion have been split now and both sections have been updated. The results section starts at line 357 and is split into major ions, water stable isotopes and isotopic tracers. The discussion section at line 438 and contains the same headings as the original paper.

**The next part of this discussion (lines 351 to 362) also mixes observations with conclusions and deals with a variety of processes (mixing, evapotranspiration, and calcite saturation). There are again few details; what does "towards calcite saturation" mean (what are the saturation indices, where do you discuss them)? Why is the calcrete important (is it found at groundwater discharge points in the watershed, for example?).**

The discussion on calcrete has been deleted in the streamlining of the manuscript in our attempt to remove anything that was tangential to the primary story we are presenting.

**I agree that the halogen chemistry in Fig. 4 is consistent with evaporation but you need to explain why that is so in the text (perhaps more to the point is this important for understanding recharge and mixing?).**

At line 453 as part of the new discussion we state "As we assume that Cl⁻ is behaving conservatively (Appelo & Postma 2005) we surmise that increases in dissolved major ion concentrations concomitant with increases in Cl⁻ in the shallow groundwater are likely to be a result of evaporation" to explain our interpretations regarding the increases in major ion chemistry. Additionally, this comment now refers to the new Figure 3a, b, and c in the results section. Further, all parts of figure 3 are also primarily used to show mixing between the GAB and the alluvial groundwater.

**Most of the other sections / paragraphs in the results and discussion are similarly hard to follow. For example, the discussion of Tritium (lines 416 to 431) includes discussion on historic Tritium concentrations with the distribution of Tritium from this study. The latter is never illustrated (the data are in a supplemental table but the spatial variation is important so should go on a map or section). The Tritium activities are not specified so sentences such as "However, despite decreasing activities, 3H remains relatively prevalent in the deeper part of the system." are very non-specific. The conclusion reached "This indicates the extent of recharge from episodic flooding and shows that surface recharge reaches the deeper LNA (down to _80 m bgs) relatively quickly (< 70 years)." then becomes impossible to assess.**

3H activities are now included and described in the results section (Section 4.3 – line 417). Past data is introduced mentioned in the discussion (lines 483-488), to put our results into regional and temporal context. We have now added a plot of 3H (TU) vs depth (Figure 5 - as seen below) so that the reader can easily put into context the activity of 3H at a given depth throughout the LNA. The rest of the discussion has been edited so that any results are now in the results section and the writing cleaned up.

[Figure]

**There has been a lot of work on determining residence times and mixing from concentrations of radioactive isotopes (including by the coauthors of this paper). Much**

**more could be done to firm up the conclusions, for example some of the samples on Fig. 5 seem to not have undergone extensive mixing and probably could be used to determine residence times.**

It is correct that some samples in the old Figure 5 (now Figure 6 in the revised manuscript) have not undergone extensive mixing. That is firmly portrayed in the original figure 6 (now Figure 7) for the same dataset. We included residence time ranges in figure 6 (now 7) for both end-members. However, this is not a paper about complex dating corrections and decoupling of residence times in mixed samples, which we have now emphasised at lines 108-112. The emphasis is on identifying main groundwater sources within an alluvial aquifer, the use of a multi-tracer approach and to bring attention to water managers about the dangers of utilising simple hydrogeological models without incorporating any hydrochemical evidence.

**In section 4.3 it is not clear whether the Chlorine-36 data are being interpreted in terms of ages or mixing. Elsewhere, you have stressed mixing but here you calculate ages. This is despite Chlorine-36 being notoriously difficult to use for anything other than broad indications of residence times due to the input function varying (in unknown ways) over time due to climate variations. The Chlorine-36 ages are presented without much scepticism or discussion.**

We agree that $36Cl$ can be notoriously difficult to interpret and we have acknowledged this in text now (lines 87-90). However, the Pilliga Sandstone has low and fairly consistent Cl concentrations, so if there was ever a system where $36Cl$ could be used to say something about GAB residence times, it is this one (this has also been mentioned at the same location: lines 87-90). This is the same for the Coonamble Embayment elsewhere (eg. Mahara et al. 2007). There is an abundance of regional background information on $36Cl$, which we refer to throughout the manuscript. Using similar assumptions applied to $36Cl$ elsewhere in the GAB, we provide some residence times (line 520), only applicable for the extreme GAB end-member. The caution and scepticism regarding "absolute ages" is incorporated in figure 6 (now Figure 7). The $36Cl$ section (section 5.2 – line 506) reinforces the mixing processes and only provides some residence time interpretation for the extreme end-members, similarly to what is done with $3H$ (see original figure 6).

Mahara, Y., Habermehl, M.A., Miyakawa, K., Shimada, J. and Mizuochi, Y. (2007) Can the 4He clock be calibrated by 36Cl for groundwater dating? Nuclear Instruments and Methods in Physics Research Section B: Beam Interactions with Materials and Atoms 259, 536-546.

**Calculations through Section 4 are poorly presented. For example, the discussion of mixing (lines 509-530) uses a single composition for the end-members and these is no sensitivity analysis. The results are presented without much discussion of uncertainties etc.**

We have added a 'Geochemical calculations' section (section 3.3 – line 326) to explain our calculations in the old results and discussion. Regarding uncertainties, mixing and other processes (such as evapotranspiration and irrigation returns) can overprint the simple Cl mixing. However, the Cl mixing isn't considered in isolation – this is where our multiple tracers come into effect. For example, samples with [Cl] < 31 mg/L, 3H activities above our detection limit (> 0.04 TU) and/or 14C > 90 pmc are considered as 100% modern (< 70 a) flood recharge (eg. 36001-1&2, 25329-1, 30345-1, 25332-1, 25327-1&2). In this case, two samples (25327-1&2) with anomalously high $NO_3$ concentrations could be influenced by potential irrigation returns. Conversely, samples recovered from generally deeper piezometers along the B-B' section in figure 2 have higher Cl concentration that coincide with 3H below or very close to the detection limit, 14C contents generally < 5 pmc and, most importantly, 36Cl/Cl below 58 ($\times 10^{-15}$) and are considered to be comprised of a higher percentage of GAB groundwater.

**Section 4 is not helped by its structure. It mixes introductory material (e.g., lines 453-455), conclusions, and description. It is also not very tightly written and uses mainly qualitative descriptors rather than specific values. It is possible to mix results and discussion, but it needs skill otherwise the text becomes meandering and there is commonly not the rigor in explaining the salient features of the data before they are interpreted. The conclusions of this paper are plausible, but the way that it is written does not do them justice.**

The results and discussion have now been split and the writing throughout the entire paper tightened. All the results are now explained using their specific numbers in section 4 and we have aimed to remove most of the qualitative descriptors.

**Some suggestions for revamping the paper are:**
**1) Separate the results and discussion and make sure that you adequately describe the important pieces of the data (don't just say which diagram or figure it is contained in).**

Results and discussion have been separated. The results section now describes important aspects of the data such as parameter ranges and trends (Section 4).

**2) Concentrate on what is important. The aims of this study is to understand recharge and mixing, in which case some of the details of the water chemistry seem superfluous. For example, it is important to determine whether the GAB waters and local recharge have different compositions but some of the details of the processes could be omitted. The Chlorine-36 is more valuable as an indicator of mixing rather than residence time, but recharge rates estimated from Tritium are important.**

The manuscript has been cleaned up substantially, primarily as a result of the restructure. Discussions on sustainability have been omitted and the geochemistry of the groundwater, and our multi-tracer approach to estimating GAB discharge made the focus. We have further included clarification of the 3H results (Figure 5 and section 5.1.1 – line 476) and detail on how we use the 36Cl data (section 3.3 – line 326, section 5.2 - 506, Figure 2).

**3) In a similar vein, the paper would be improved by more hydrologic information at the expense of some of the detailed geologic information. Is the interpretation of mixing consistent with the hydraulic heads? What are the groundwater flowpaths?**

The geologic information has been considerably shortened and we have added more hydrologic information such as the general groundwater flowpaths and K values (lines 180-184).

**4) Include enough justification of calculations to make them convincing and some sensitivity analysis or discussion of uncertainties.**

Please refer to our comments above regarding the calculations.

**I am guessing that the senior author is a graduate student. The coauthors, however, are not and should have picked up on the more obvious problems with the way that this study was framed and presented.**

**Specific comments**
**Introduction**
**The introduction provides a general outline of the science and the reasons for carrying out the study. The first paragraph is not very clearly expressed. For example, Why specify "modern infiltration" – recharge implies modern processes**

The first paragraph has been restructured and edited. "modern" has been removed

**"Spatial and temporal data resolution and heterogeneity in hydrogeological properties result in considerable uncertainty when allocating recharge to each source and mapping pathways of flow" is very unclear.**

At lines 53-54 this sentence has been revised to make it more comprehensible. "…mapping pathways of flow" has been removed as it wasn't pertinent to the story (compared to allocating sources of recharge) and made the sentence laborious.

**What is a "dynamic groundwater gradient"?**

"Dynamic groundwater gradient" has been changed to "…change in the magnitude of groundwater gradients and direction over time…" (line 56).

**The introductions to papers are important as they frame the study and hopefully persuade the reader to continue reading, so it is worth making them as clear as possible. Line 86. What do you mean by "modern/submodern"?**

Submodern has been removed.

**Line 89. It would be useful to explain briefly how the various isotopes help understand mixing as it might not be clear to all readers.**

We have included a simple sentence at lines 77-79 to explain how various isotopes help understand mixing ("Isotopes of dissolved species can be useful for elucidating groundwater mixing provided the different sources of groundwater have distinctly different and consistent isotopic signatures"). We also refer to Bentley et al. 1986; Andrews & Fontes 1993; Love et al. 2000; Moya et al. 2016 at this location for the reader to peruse for further information.

**Line 96. It would be clearer if you split this material off as your objectives get lost at the end of the discussion of the techniques (perhaps put a subheading in for emphasis).**

We have made this a new paragraph so as to emphasise the objectives (lines 97-109).

**Lines 98-99. This is stating a conclusion, which you should leave until later.**

We have changed the end of the introduction (lines 97-109) to now state only what we did, rather than what we found and moved the conclusions to the end of the manuscript.

**Study Area**
**This section provides a comprehensive description of the study area. Some specific comments**
**Lines 139-141 is difficult to follow for anyone not familiar with eastern Australian groundwater hydrology. Can you add a key map of the basins to Fig. 2?**

We have simplified this text so that it is not quite so laborious on the reader and simplifies the concept we were trying to convey. We have also referred to our main GAB reference (Radke et al. 2000), which has some excellent basin maps for the readers perusal. We have not included a map in Figure 2 of this work because we did not want to oversaturate with figures and have too many geological maps in the one figure.

**Lines 141-162. The description of geological framework is difficult to relate to Fig. 2 as you do not specify the age of the various units. Provide a few more details in the text.**

Age of the various units has been specified in text throughout lines 141-156 now.

**Section 2.1 is probably too detailed for the study. While understanding the geology of the area is important, I am not convinced that the geologic history needs to be gone through in this much detail (for example, is the rainfall variation between the Miocene and today important for this study). This section could be cut fairly substantially. What would be more useful, and which is not there, are firstly some hydraulic properties (K, porosity etc) and secondly some description of groundwater flow.**

This section has been shortened considerably. There is no porosity or permeability data for the region, however we have added some K values and we explain the general direction of groundwater flow (lines 180-184).

**Lines 197-201. This repeats material in the introduction and could be removed.**

This section at 188-192 has been shortened, however not removed completely because it highlights the gap in the literature regarding the use of catchment water balance models together with hydrochemical data.

**Line 205. The numerous Merrick references seem to be to a series of non-publically available documents. The reference to the Kelly et al., summary would seem sufficient.**

The references have been eliminated and instead Merrick 2000, a PhD thesis available through the UTS library, has been used (line 196). Additionally, Kelly et al. 2007 has been included in many of these places.

**Lines 208-209. Not clear what you mean by "There are equivalent solutions for all water balance models and the solution presented is often constrained by several factors."**

We agree and this sentence has been changed to "There are multiple plausible solutions for all…" (line 198).

**Section 2.2.1 does not add that much useful. You state that there are a range of models but provide few details. The discussion of the models seems to reside only in unavailable**

**consultants' reports and then the point about not taking into account geochemistry is reiterated. This section could be shortened, especially as you do not make detailed comparisons with specific models later in the paper.**

This section is core to the problem with respect to the issue of water balance modelling in the Namoi. We have removed the references to unavailable reports and replaced with a PhD thesis (Merrick 2000) but we have left the section (beginning at line 194) in the manuscript so that the 22% GAB contribution that we later use for comparison to our estimates of 70% have some basis.

**Figures**
**Fig. 1. Define "bgs". Rather than describing what is on the two axes, just label them in the graphs. Text on the map is too small (you could make the map larger and put the Australian map in an unused corner)**

Bgs has been defined. We describe the axes rather than place them on the graphs because the font is too small otherwise and the repetition of labelling the axes on every graph was taking up too much room. The map has been made larger and the map of Australia placed in an unused corner of the map.

**General Comments:** **This manuscript has potential to be a very interesting article and could have a big impact on our current understanding of the interactions between alluvial aquifers, regional aquifers, and surface-water systems. The topic should have broad interest to other research on alluvial aquifers and broad applicability to global alluvial groundwater systems. The combined geochemical tracer and multiple isotopic tracer approach is a potentially robust way to sort out these interactions. However, the manuscript suffers from poor organization and lack of specificity in the methods and interpretations. This prevents a recommendation to accept the article as it is currently written. Instead, the recommendation is to accept after major revisions including the organization and technical detail of the article. In addition, it was very difficult to read the manuscript due to awkward syntax and lack of focus (see Specific Comments).**

We thank the reviewer for their time in reviewing our manuscript closely. We have attempted to address all concerns below, particularly the major concern of the organisation and technical detail of the article and the syntax and focus. The manuscript has been restructured and this has made the manuscript more focused on the key messages.

**The major findings/contributions of this manuscript hinge on the authors' interpretations of the isotopic and geochemical data. In general, these sections need much more clarification than is provided and in some cases, additional data mining may be necessary (although I think the latter is the smaller of the issues). For example, a considerable emphasis is placed on 3H concentrations. However, almost all of the 3H concentrations are close to background or are 3H dead. This has implications for the mixing model and it has implications for the residence time estimates. It's not unusual to find 3H dead waters which have slightly elevated 36Cl/Cl ratios. 3H, even from the bomb-pulse, is decaying faster than 36Cl even in the presence of mixing with recent recharge and sometimes 36Cl does a better job of sorting out mixing than 3H (especially for waters that are 100 years or older). However, the real strength that the authors have is the ability to sort out young and old fractions (relative terminology) using 14C pmc. In the absence of carbonate exchange and mixing in the aquifer (another topic which needs additional clarification) the 14C pmc value is very useful in sorting out these endmembers as the authors attempt to do in Figures 5 and 6. In any case, the authors need to provide additional clarification on how they sorted out the mixing processes that affect these tracers.**

**Regarding data mining, this manuscript and the authors' interpretations would be strengthened by providing the 3H, 14C, and 36Cl/Cl of modern precipitation. Are these data available? If so, cite them. This would enhance the mixing model which is used to determine the fractions of GAB water in the LNA. This calculation also needs additional clarification and the authors should present the equation and discuss sources/estimates of uncertainty in this calculation.**

These are some interesting observations that we have endeavoured to address here and in the re-organised manuscript. Firstly, we would like to stress a couple of points specific to our data and the southern hemisphere that we will emphasise in the revised manuscript (lines 422-425). The quantification limit for 3H in our lab is 0.04 TU, so only 16 out of 50 samples are either at the detection limit or below it. This low detection limit has been driven by low 3H concentrations in the southern hemisphere. At the peak of the bomb pulse, weighted averages in Australia were around 60 TU (compared to, for example, ~ 6000 TU in Ottawa). In Australia, tritium in rainfall has been at natural background levels for some time. Values from samples in the Namoi River, collected at the time of our groundwater sampling (and provided in the dataset: ~2.3 TU) are in good agreement with rainfall data provided in Tadros et al. 2014.

Regarding 14C, whilst not generally sampled in rain or surface samples, it is almost at natural/background content, as seen in the Namoi River samples (~102 pmc; also provided in the dataset). However, shallow groundwater, recharged in recent years, still has a slight bomb pulse component in some boreholes (values of around 107 pmc).

Regarding 36Cl/Cl, no data in rainfall exist in the region to our knowledge. Namoi River samples at two different times (summer and winter 2016) have values of ~416 ($x10^{-15}$), while tributaries draining the igneous rock terrains have higher ratios than modern river water. However, most shallow groundwater also shows the mixing with deeper 36Cl/Cl sources and therefore has lower 36Cl/Cl ratios than modern surface waters.

We have added a 'Geochemical calculations' section to the methods (section 3.3) that details the method used for the calculations in old Section 4 (now section 5 in the rearranged manuscript). In terms of sources of uncertainty in this calculation, mixing and other processes (such as evapotranspiration and irrigation returns) can overprint the simple Cl mixing. However, we draw our interpretations from our Cl mixing results, alongside the other tracers that we used. For example, we consider samples with [Cl] < 31 mg/L, 3H activities above our detection limit and/or 14C > 90 pmc as originating from 100% modern (< 70 a) flood recharge (eg 36001-1&2, 25329-1, 30345-1. 25332-1. 25327-1&2). Samples recovered from generally deeper piezometers along the B-B' section in figure 2 show higher [Cl] that coincide with 3H below or very close to detection limits, 14C contents generally < 5 pmc and, most importantly, 36Cl/Cl below 58 ($x10^{-15}$).

**Specific Comments:**
**Line 56: This paragraph lacks focus. The authors bring in agriculture and it's not clear how this is connected to the bigger issue.**

This paragraph has been edited to try to bring it into focus. Agriculture is no longer mentioned here, just groundwater abstraction for irrigation, stock and domestic water supplies (lines 57-58).

**Line 64: What is meant by the international export market?**

This part of the introduction has been removed.

**Lines 65-78: What is the focus here? Aquifers in general or alluvial aquifers?**

Alluvial aquifers are the focus. This has been clarified in text at line 62.

**Line 81: It's not just the half-life that is important, the tracer systematics including mixing and processes that affect their interpretations are needed.**

We have acknowledged that each tracer undergoes processes specific to it that can often affect the interpretation of it and therefore these tracers can provide insights into the groundwater, but for only a given window of time and in favourable conditions (lines 80-81).

**Lines 96-105: The manuscript needs a focused statement about the current knowledge gaps (this should be developed in the Introduction more succinctly) and what is new and novel about this research in addressing those gaps. The manuscript would benefit from a clear hypothesis statement or statement of science questions. This, in my opinion, would help focus the entire manuscript and the authors should return to this statement in the opening paragraph of the conclusions.**

The introduction has been edited and aspects of it cut. The last paragraph (lines 97-114) has been edited to highlight that we are for the first time (to our knowledge) combining multiple geochemical tracers to assess artesian discharge to an alluvial aquifer and that we show a need to combine this with water balance models of alluvial systems. This is reiterated in the conclusions.

**Lines 208-213: This sentence is difficult to understand, yet this paragraph is critical in identifying the knowledge gaps.**

This sentence has been changed to "..there are multiple plausible solutions for all water balance models…" at lines 198-199

**Line 237: This is a good place to re-state or reiterate what is missing by identifying the knowledge gaps and how your research addresses those gaps.**

We have moved a section of the introduction to here (lines 226-230), stating that there is an over-reliance on water balance models, which are not constrained by hydrochemistry and that this study shows that the conceptual insights from geochemistry provide a new set of constraints not considered in previous hydrogeological models.

**Lines 292-293: Please clarify the statement on NH4 concentrations.**

A sentence has been added to clarify the statement on NH4 concentration (lines 333-335). We explain that the NH4 concentration skewed the charge balance because it was not considered in the initial calculations. This is now in section 3.3 – geochemical calculations.

**Line 336: Does it make more sense to separate the Results and Discussion. This may help streamline and better organize the manuscript.**

The results and discussion have now been split and the discussion streamlined.

**Lines 343-362: This needs to be better organized, perhaps start with description of how GAB works hydrologically, then describe the ratios and their implications for the LNA. Suggest breaking this paragraph into 2 new, concise paragraphs.**

The original lines 343-362 have been changed because of the restructure. Some of the original text is now in section 5.1, line 449, but the organisation has been changed considerably.

**Lines 375-376: Be assertive here. This does suggest...rather than may suggest.**

We have changed to 'This suggests…' (line 520).

**Lines 376-378: Needs clarification.**

This has been split up because of the split in results and discussion. Section 4.2 details the ranges of the isotopic values and that they lie on an evaporation line whereas lines 443-448 now explain the connection between evaporated surface water and shallow groundwater.

**Lines 380-381: Please provide sources of F- and how that relates to the point you are making.**

The source of F⁻ is the weathering of silicate minerals in the region (noted now at lines 460-461). This relates to our overall message because the covariation of Na and F (from the weathering of silicate minerals) show the same mixing trend in the alluvium between shallow groundwater and GAB samples that other parameters do (such as Na-HCO3).

**Lines 416-431: Please clarify. Most 3H values are close to background or are dead with respect to 3H. This complicates your interpretation, but you seem to pull it back in focus with the figure. Suggest picking specific sites and describe what the data is telling you.**

Please refer to our previous comments above regarding the activity of 3H in the southern hemisphere.

**Lines 432-434: Prevalence of 3H?? Again, the 3H values are almost all very low or 3H dead. Consider rewording and clarifying this statement. Also, are there recent 3H values for precipitation?**

"Prevalence of 3H" has been changed to "the activities of 3H above the detection limit" (line 492) so as to be more precise with our wording. Please refer to our previous comment regarding 3H activities in Aus and values for precipitation.

**Line 434: 3H and 14C and not entirely consistent are they? This needs clarification. Why are they inconsistent?**

Figure 5 (now Figure 6 in revised manuscript) illustrates why the 3H and 14C in the groundwater are inconsistent. Normally we would expect to see a very sharp decline in 3H activities as 14C (pmc) decreases. However, we see a mixing trend that shows that some groundwater samples still have measurable 3H despite having a 14C content of less than 50 pmc. We have clarified this in text at lines 492-503, better explaining the figure.

**Lines 432-445: Please clarify and provide a more concise discussion on mixing effects.**

This section (now lines 492-503) has been clarified with respect to the 3H (see above comment). It has also been shortened as a result of the streamlining of tangential information in the manuscript (mention of calcrete removed).

**Line 458: What about 36Cl/Cl of modern precipitation? High 36Cl/Cl can be indicative of mixing of bomb-pulse with recent (low 36Cl/Cl) recharge. But, low 36Cl/Cl can also imply very old groundwater (your case). Can you clarify this uncertainty with modern precipitation?**

Please refer to our comments above concerning 36Cl/Cl values of modern precipitation. In regards to lingering 36Cl/Cl derived from the bomb pulse: Cl has a very low residence time in the atmosphere (2 days) so most of the bomb derived 36Cl would be gone.

**Line 487 (Figure 7): Please cite Phillips (2000) Chapter 10.**

This has been cited (line 535) and added to the reference list.

**Lines 510-523: Please provide the equation used to calculate the mixing proportions. Can you also provide estimates of uncertainty in these calculations? Are there other solutes or solute ratios (Cl/Br for example) which may be more suitable for these calculations? It's not clear how this was estimated.**

Mixing proportions were calculated in NetPath and were treated as a mixing problem. We have added a 'geochemical calculations' section to the methods that explains this (section 3.3). Please refer to our comment above regarding mixing proportions. Additionally, we believe that the Cl ion was the most appropriate because of its conservative nature (line 556) and in our study area the end member water sources have distinct Cl signatures that facilitate its use as a tracer.

**Line 535: Can you provide plots showing the spatial map of appropriate chemical concentrations?**

Figure 2 has now been adapted as below. We show a spatial map of the 36Cl data, which is crucial for understanding the discussion regarding GAB discharge into the alluvium at lines 433-439.

[Figure]

**General comments:** **The presented study is well designed and informative for regions, where different water bodies seem to exist and mix in ratios, which are unknown yet. Thematically the paper fits to HESS, although I see some points of weakness, mainly related to formulation (or omitting) of hard facts. I guess, with considerable revision, that manuscript has the potential to be of interest for a wide audience. In general, the manuscript should be shortened and particularly the geological part must be clarified for readers outside Australia. In the following, I give some specific remarks to points, where I see difficulties:**

We thank the reviewer for their time in closely reviewing our manuscript. We have aimed to address all concerns below.

**Hydrographs in Figure 1 are not very informative, despite the information, that gw-tables are fluctuating. the legends of hydrographs are not explained and it becomes not obvious, why red texted hydrographs are representative for GAB contribution. Instead of showing relative depths of screen bottoms (bgs), it would be more distinctive, when depths would be given relative to msl., to explain the absolute depth.**

We have now explained in the legend of Figure 1 why the 2 red text hydrographs show GAB contribution (the deeper piezometric head is higher than head from the shallower wells, indicating upward flow). We think that the hydrographs figure gives the reader a nice overview of the state of the groundwater system in the study area, in particular, areas that are affected by seasonal pumping and areas where the GAB may be contributing water. The NSW Government hydrograph data available only lists the below ground surface depth, however in the region that the hydrographs cover, there is very little elevation change (80 m change in elevation from the south-east to the north-west portions of the study area).

**Hydrogeological setting The entire paragraph is very hard to understand, since local formation names are abundant and the hydrogeological context is not clear. Why are all these details necessary for the reader of the manuscript (e.g. lines 191-193)? Paleogeographic features are very difficult to understand. It would be of more importance to reduce the (doubtless interesting) geological context and focus on the formations, which are hydraulically relevant. Probably a stratigraphic table would help a lot, showing thickness, lithological composition and phreatic/confined conditions in each of the relevant formations.**

The geological section has been shortened, with most of the second half (including old lines 191-193) removed. We have not included a stratigraphic table, only because the legend in Figure 2 shows a stratigraphic column by age and the cross sections in figure 2 show the depth of the different formations in the Jurassic and Triassic. However, we have cleaned the section up substantially (including removing most paleogeographic features) and have included some 36Cl data in the cross sections of Figure 2.

**221 Water balance modeling for recharge That paragraph explains a series of MODFLOW attempts to define various sources for recharge. I believe, the paragraph is to long, since the basic and neccessary information are the outcoming numbers (ratios) for the different proposed sources. The authors use a unit (ML/a) which is unknown to me (Megalitres/year?)**

This paragraph has undergone minor phrasing edits (eg line 198: "multiple plausible solutions"

has replaced "there are numerous equivalent solutions…"); however, the bulk remains because it is core to the problem with respect to the issue of water balance modelling in the Namoi. ML/a is megalitres/year. We aimed to be consistent using "a" (annum) for year throughout the manuscript, because we referred to "ka" (kilo annum) when assessing the 36Cl residence times.

**Specific comments**
**Line 302: what is the reason to use pmc and pMC?**

We detailed both units (normalised pMC and de-normalised pmc) for completeness and to ensure our results were comparable with other study results reported only as pMC's without having to transform units or search information relating to the type of AMS used, or how the d13CDIC was obtained. However, the text has all data consistently as pmc following best practice for reporting groundwater 14C data (Plummer & Glynn 2013). pMC results are only presented in the Supplementary Table.

**Line 358: which 2 processes are meant? ET leads to enrichment of all elements, leading eventually to Cc-saturation. Na/HCO3 increases only, when calcite precipitates.**

This sentence has been removed now as part of streamlining the manuscript. Additionally, at lines 456-458 (where this old sentence would have been in the restructure) we have explained why we consider an evaporative enrichment in all elements.

**Line 360: I suggest to be careful in interpreting Cl/Br ratio changes in these context. Cl/Br ratio will change only, when degree of evaporation results in supersaturation of the water in respect to halite, otherwise there is no change observable. Since Cc-precipitation is discussed, it might be worthwhile to compare Ca/Mg ratios and (Ca+Mg)/HCO3 ratios? Cl/Br ratio might change due to geological reasons...**

This is an omission on our part. We should have mentioned the Cl/Br ratio in the context of the Cl v Cl/Br plot. It is not the Cl/Br ratio itself, but rather the trend when plotted against Cl that provides the evidence for our claim. However, in the restructure of the manuscript the original sentence at 358 ("the evaporative enrichment is also evident in the concentrations of F, Cl and Cl/Br") has been removed. We have instead made these data (presented in the new Figure 3) more relevant to the main groundwater mixing message that this manuscript conveys, rather than tangential processes. We had a look at Ca/Mg and (Ca+Mg)/HCO ratios but they revealed nothing additional.

**line 390: delete charges. What means "closer"? compared to what?**

Charges have been deleted. "Closer to seawater…" has been changed "more similar to seawater…" at line 382.

**line 399-401: that sentence is not helpful, since the reader does not know which parameters you refer to. From Figs 3 and 4 it is not given, that 273314 resembles river water, it is obviously just fresh water.**

This sentence has been removed as the parameters concerning the outlying sample are described in the separate results section at lines 383-399 (particularly line 397) now.

**line 401 ff: from that moment it becomes highly difficult to follow: you refer to the only**

**sample from the Jurassic Fmt. Why is it strange to have fresh water in there? The base of the well is just above a Napperby fmt. Which indicators suggest recharge through a formation, which is even below Napperby? And which river is referred to? Why should Pilliga Sandstone contribute? The explanation lacks from facts, which give an overview about the hydraulic concept, which obviously led to the formulations. Latest here a regional W-E geological cross-section, showing Fmts. of GAB and their regional confined and phreatic conditions (piezometer heights) is urgently needed to understand the hydrogeological context of the region. In addition, it would also help, to (i) show Fmt. and (ii) add water table heights of the different aquifers in the cross-sections of Fig. 2. Situation becomes harder due to the jumping between formation names.**

It is an anomaly to have such fresh water in this sample because the well screen is 207 m bgs and in the GAB Jurassic formation. All other studies that have taken samples at this depth in the GAB have had water that is more representative of the GAB, ie higher TDS. We don't have particular indicators that suggest recharge through the Digby into the Napperby, however this was our hypothesis because of the fresh water 207 m bgs, and the contact between the Namoi River and the Digby Fmt to the south of our study area.

The Namoi River is referred to. This is the only river in our study area and it is in direct contact at the surface with the Digby Fmt to the south of our specific study area. The Pilliga Sandstone should contribute because the sample is taken from the Pilliga SS. "Overlying Pilliga SS" has been changed to "Pilliga SS" to try to make that clearer (line 472).

We have changed 'This suggests' to 'We hypothesise' (line 469) because the first sentence did not necessarily suggest the second.

Figure 2 provides a cross section of the geology and piezometer height, as well as a geological map that shows the Digby Fmt outcropping to the south of the study area, where it is in contact with the Namoi River.

The formations are shown in the cross section of figure 2. Water table heights change over time due to floods and the intensive pumping in the study area. We have included the standing water level at the time of sampling in the Supplementary Information.

A paragraph at lines 467-477 now aims to describe all the above better.

**line 407 ff: again, why do the authors claim for contact between that river and deeper Triassic Fmt.? According to Fig. 2: Napperby is the uppermost Triassic. Where is that river situated and why is the river the only option of fresh-water supply? Are these ideas consistent with hydraulic?**

At the depth that these samples are situated at, and given the regional data for the quality of the GAB groundwater, the river is the primary source of fresh water. We claim contact between the river and deeper Triassic formations because the river contacts the deeper Digby to the east of the study site, which is consistent with the groundwater flowpaths in the region. This then is important because deeper mixing between the GAB groundwater and the water of the Namoi River is an important consideration in water balance models of the catchment. This is all clarified at lines 467-477 now.

**Line 412f. : To be very critically: I don't see clear indications for that statement from Figs. 3 and 4. Major elements in samples >80 m (blue) spread over the entire range and only a few blue samples fall in the same region as GAB analyses from Radke et al. (2000). Is there a geographic link?**

We interpret the spread of samples from the shallow, intermediate and deep, and the mixing between them over the entire range of Figs 3 and 4 to be indicative of the mixing occurring throughout the groundwater system. Most of the blue samples in new Figures 3 and 4 fall in the same region as GAB samples. That the deeper blue samples cluster more with the GAB samples than the red or green indicates to us that the deeper groundwater is experiencing greater mixing with the GAB in some places, which is then mixing in varying proportions with the more shallow groundwater, up until the very shallow groundwater is comprised entirely of itself at the other end of the mixing trend. We acknowledge that the blue (deep) samples are spread over a large range, this to us indicates significant mixing throughout the entire vertical groundwater profile. There is no specific geographic link that we have been able to identify.

**Line 517f. : Why is not a sample chosen, which was not evaporated at all or even better, a recent rainfall sample, giving the precise input signal for Cl and 3H?**

Choosing a more evaporated end-member makes our calculations even more conservative, as it ensures we aren't overestimating the GAB contribution. Rainfall and recharged shallow groundwater do not usually have the same Cl composition, generally Cl is higher in the shallow groundwater. Additionally, we chose the groundwater sample with the highest $^3$H activity, as an indicator of recent surface infiltration.

**lines 521-523: I do not understand the reason of that thought: "...to consider overall transport of Cl from shallow groundwater."**

This sentence has been changed to "Thus, the use of the evaporated sample as our end-member represents a conservative approach when considering the mixing components from both the LNA and the GAB" (line 567-569) so that it is clearer.

**line 532/fig. 8: Actually these percentages are calculated on Cl-mixing approach only. Within the description, "multiple geochemical tracers and major ion data" are mentioned. Which exactly were used and how does the respective results fit to the described Cl-mixing? According to that figure, it strikes, that heterogeneity of GAB contribution might be related to structural features or any other elements that provide preferential flow? Are there any tectonic lineaments or other indications, which could be responsible for the different contributions from the GAB?**

Cl- only was used for this percentage based assessment of GAB contribution, but the other major tracers were used to qualitatively show GAB contribution to the alluvial groundwater. We used the multiple tracers approach to constrain GAB discharge and used the Cl ion to estimate a percentage. For example, we consider samples with [Cl] < 31 mg/L. 3H activities above our detection limit (> 0.04 TU) and/or 14C > 90 pmc to be 100% modern (< 70 a) flood recharge (for example: 36001-1&2, 25329-1, 30345-1, 25332-1, 25327-1&2). Samples that were recovered from generally deeper piezometers along the B-B' section in figure 2 show higher [Cl] that coincide with 3H below or very close to the detection limit, 14C contents generally < 5 pmc and, most importantly, 36Cl/Cl below 58 (x10$^{-15}$). There are no other indications that could be responsible for different contributions from the GAB.

**Lines 619-621: That sentence is very vegetarian, it gives no information at all. Please prevent to use such phrases, instead of describing which reason will result in which effect.**

This sentence in the conclusion has been removed.

[revised manuscript text omitted]

This has implications for ongoing groundwater use in the region, and highlights the need to protect surface recharge zones in both alluvial and artesian portions of catchments.

The over-reliance of water balance models used to allocate groundwater resources that have not been constrained by isotopic tracer residence times or hydrochemical results is a common issue globally. This research highlights that comprehensive hydrochemical investigations improve our conceptual understanding of recharge pathways and that such investigations should be applied to all important groundwater resource assessments to enable sustainable management.

[Figure]

[Figure]

**Figure 1.** Map of the study area and sample locations, along with the location of the study area in Australia. Accompanying hydrographs show the groundwater level response in different piezometers throughout the study area (groundwater level data sourced from BOM 2017). The different colours in the hydrographs represent the different monitoring bores in the nested set. The bottom of the slotted interval for each bore is shown in the key. The x-axis in each hydrograph is the year (1970-2010) and the y-axis is depth (between 0 and 40 m below ground surface (bgs)). The two locations with red text highlight areas where the hydrograph heads show clear GAB contribution, with the deeper piezometer showing a higher head than the shallow one. The remaining locations show no apparent GAB contribution to the LNA based on the hydrograph data.

**2    Study Area**

The lower Namoi River catchment is located in the north-west of NSW, Australia (Figure 1). Groundwater resources in the LNA are the most intensively developed in NSW (DPI Water 2017). For this reason, there is concern regarding groundwater exploitation and threat to the long-term sustainability of the system (Lower Namoi Groundwater 2008; DPI Water 2017). Groundwater abstraction from the LNA supports a multibillion-dollar agricultural sector (focused around cotton growing established in the 1960s), supplying around 50% of water for irrigation in the region (Powell et al. 2011). The first high volume irrigation bore was installed in 1966 (Rural Forum 1967) and the use of groundwater expanded rapidly throughout the region throughout the 1960s to 1990s. Peak extraction of approximately 170,000 mega litres (ML) occurred over the 1994/1995 growing season (Smithson 2009). Consistently declining groundwater levels and concern regarding the long-term sustainability of groundwater abstraction led to the implementation of a Water Sharing Plan in 2006, which systematically reduced groundwater allocations to the irrigation sector over a ten-year period. The present allocation is 86,000 ML/a (Lower Namoi Groundwater 2008).

**2.1    Hydrogeological setting**

The lower Namoi River catchment lies within the Murray-Darling Basin, overlying the Coonamble Embayment, which is in the south-east portion of the GAB (Radke et al. 2000). The lower Namoi River catchment lies within the Murray-Darling Basin and overlies the confined Coonamble Embayment, which is a subdivision of the Surat Basin, which in turn is a sub-basin of the GAB. The southernmost portion of the LNA is underlain by Triassic formations, while northwest of monitoring bore 30345 the LNA is underlain by Jurassic formations (Figure 2). Within the region of study, the oldest outcropping bedrock formation is the early Triassic Digby Formation (lithic and quartz conglomerates, sandstones and minor finer grained sediments) (Tadros 1993). The Digby Formation outcrops in the south-east of the area and the Namoi River abuts the formation just south of B' on Figure 2. The Digby Formation is overlain by the Triassic Napperby Formation (thinly bedded claystone, siltstones and sandstone). This formation occurs at a depth of 106 m, just below the base of monitoring bore 30345 (NSW Pinneena Groundwater Database, driller logs)., where the paleo Namoi river carved a path through a syncline. In outcrops to the east of the study area, the Napperby Formation is overlain by the late Triassic Deriah Formation (green lithic sandstone rich in volcanic fragments and mud clasts) (Tadros 1993)., however this has not been identified beneath the bores used in this study. The boundary between the Triassic and Jurassic lies west of monitoring bore 30345. There is an unconformable boundary between the Triassic and Jurassic formations, and in some outcropping regions the Garrawilla Volcanics (alkali basalts, trachyte, hawaite, pyroclastic and subordinate sediments) is the base Jurassic formation. Overlying the Garrawilla Volcanics are theThe early Jurassic formations important to this study are the Purlawaugh Formation (carbonaceous claystone, siltstone, sandstone and subordinate coal), Jurassic Pilliga Sandstone (medium to coarse quartzose sandstone; Tadros 1993) and the late Jurassic Orallo Formation (clayey to quartzose sandstone, subordinate siltstone and conglomerate) (Tadros 1993). The Pilliga Sandstone forms the bedrock below monitoring bores 25325 to 25342, and in the Namoi region is the primary aquifer of the GAB.

[Figure]

[Figure]

**Figure 2.** Two cross sections through the study area, showing the location and depth of the samples in the alluvium and their proximity to formations of the GAB. Contacts obtained from gas wells Nyora-1, Culgoora-1 and Turrawan-2, coinciding with our cross sections, are added. Their locations are displayed on the map. The chlorine-36 data interpolated using the 'natural neighbours' algorithm is shown in each cross section.

From the late Cretaceous to the mid Miocene, a palaeovalley was carved through the basement rocks (Kelly et al. 2014). Then from the mid Miocene until present, the palaeovalley was filled with reworked alluvial sediments. Groundwater abstraction in the study area is mostly from these alluvial sediments. Fluvial and aeolian interbedded clays, silts, sands and gravels form the up to ~ 140 m thick alluvial sequence of the Lower Namoi Catchment (Williams et al. 1989). Traditionally, three main non-formally defined aquifers/formations have been used to describe the LNA. The semi-confined Cubbaroo Formation overlies the bedrock in the northern palaeochannel (which passes beneath monitoring bores 25325 and 30092). This formation is up to 60 m thick. The Cubbaroo Formation is overlain by the semi-confined Gunnedah Formation, which is up to 80 m thick, and is conformably overlain by the unconfined Narrabri Formation, which is 10 to 40 m thick (Williams et al. 1989). However,  recent  studies in the Namoi Catchment suggest that the rigid subdivision in to the Narrabri, Gunnedah, and Cubbaroo formations  cannot easily explain the continuum in chemical evolution observed (discussed further below) and that the valley filling sequence is better characterised as a distributive fluvial system  (Kelly et al. 2014, Acworth et al. 2015).

Groundwater drains from the Upper Namoi into the LNA via a bedrock constriction north of Narrabri and generally flows from east to west within the LNA (Barrett 2012). Hydraulic conductivity in the alluvial aquifer is highly variable (0.008-31 m/day) due to the presence of variable sand and clay (Golder Associates 2010). However,  hydraulic conductivity generally increases with depth. ~~Kelly et al. (2014) argue that the sedimentary sequence is better represented as a distributive fluvial system, with high energy sedimentary gravel and sand deposits dominating at depth and low energy silt and clay deposition dominating near the ground surface. This is due to a shift from a relatively wet climate in the mid Miocene (greater than 1500 mm annual rainfall; Martin 2006) to the present drier climate in the region, which averages approximately 660 mm at Narrabri (BOM). There is also a higher proportion of gravel and sand deposits in the proximal portion of the catchment, between Narrabri and Wee Waa (the area of this study), than the distal portion of the system west of Cryon (Kelly et al. 2014). Acworth et al. (2015) showed that within the alluvial sequence there are time gaps of hundreds of thousands to millions of years in the sedimentary sequence, which is expected in meandering river sedimentary environments.~~

[revised manuscript text omitted]
 in shallow groundwater samples near the main channels show areas with modern recharge. However, despite decreasing activities, ³H remains relatively prevalent in the deeper part of the system. This indicates the extent of recharge from episodic flooding and shows that surface recharge reaches the deeper LNA (down to ~80 m bgs) relatively quickly (< 70 years). In February 1971, the region experienced the second largest flood on record. Pre-flood sampling of deep groundwater (> 70 m bgs) revealed ³H activities ranging from 7.9 to 11.2 TU, in several bores located in the north of our study area (Calf 1978). The same monitoring bores in September 1971 and March 1972 ranged from 16.6 to 20.7 TU, with surface water in the Namoi River ranging from 16.9 to 22.3 TU (Calf 1978). Pre-flooding ³H activities suggest that modern water was already present in deeper parts of the alluvial aquifer at this time, indicating good connectivity to the surface and that substantial recharge took place during this flood, highlighting the importance of surface water recharge to the LNA. It should be noted that the high ³H values in the 1970s are a result of atmospheric nuclear bomb testing and can't be compared with present day ³H values. The $^{14}$C content in the groundwater ranged from 0.2 pmc to 107.6 pmc (average: 54.0 pmc). Generally, groundwater samples shallower than 30 m had a high $^{14}$C content (> 90 pmc), which decreased with depth. There were 9 samples with a $^{14}$C content below 1 pmc, indicating very old groundwater (> 30 ka), with total depths ranging from 35 m bgs to 207 m bgs.

It has been found that groundwater in the GAB recharge zone closest to the study area has a $^{36}$Cl/Cl ratio up to ~ 200 (x10$^{-15}$) (Radke et al. 2000) with recharge values applied in calculations elsewhere in the GAB of 110 (x10$^{-15}$) (Moya et al. 2016). Water from the Namoi River has a $^{36}$Cl/Cl ratio of ~ 420 (x10$^{-15}$), possibly affected by thermonuclear $^{36}$Cl input from atmospheric bomb testing in the 1950s (Supplementary Table 4). We calculated $^{36}$Cl ages residence times from the equations of Bentley et al. (1986), assuming no other sources or sinks besides recharge and natural decay (eqn. 1):

$$t = \frac{-1}{\gamma_{36}} \ln \frac{R - R_{se}}{R_0 - R_{se}} \qquad (1)$$

where R = $^{36}$Cl/Cl ratio measured in the sample, R$_0$ = the initial $^{36}$Cl/Cl ratio (meteoric water), and R$_{se}$ = the $^{36}$Cl/Cl ratio under secular equilibrium (in this case the $^{36}$Cl/Cl ratio from the Pilliga Sandstone). We used a R$_0$ value of 160 (x10$^{-15}$), which was an average of 10 samples compiled from studies in the Coonamble Embayment and reported in Radke et al. (2000). For R$_{se}$ we used a value of 5.7 (x10$^{-15}$), which is appropriate for aquifers dominated by sandstone (this secular equilibrium value can vary according to the dominant lithology). This R$_{se}$ value has been applied to $^{36}$Cl/Cl calculations elsewhere in the GAB (Moya et al. 2016).

Our $^{36}$Cl calculations results resulted in residence times for the alluvial groundwater ranged ing from 24.06 (x10$^{-15}$) to 455.35 (x10$^{-15}$) (average: 169.4 (x10$^{-15}$) (shown in the interpolation in Figure 2). It has been found that groundwater in the GAB recharge zone closest to the study area has a $^{36}$Cl/Cl ratio up to ~ 200 (x10$^{-15}$) (Radke et al. 2000) with recharge values applied in calculations elsewhere in the GAB of 110 (x10$^{-15}$) (Moya et al. 2016). Water from the Namoi River has a $^{36}$Cl/Cl ratio of ~ 420 (x10$^{-15}$), possibly affected by thermonuclear $^{36}$Cl input from atmospheric bomb testing in the 1950s (Supplementary Table 4).

**5    Discussion**

**5.1    Identification of recharge and mixing between the GAB and the LNA**

~~In the literature, mechanisms of recharge to the LNA are generally agreed upon, with a main surface water recharge component and a minor artesian component (Calf 1978; Merrick 2000; McLean 2003). We observe these two mechanisms in this study as well, however the relative contributions of these two components at any given time, and how this contribution changes over time, are difficult to constrain.groundwaterandmodern~~ surface water that has undergone evaporation and shallow groundwater.

 Additional evidence for these two mechanisms of recharge is the composition of $Na^+$ and $HCO_3^-$ in the LNA. ~~suggests the 2 aforementioned mechanisms of recharge. The Na HCO$_3$ ratio in GAB groundwater is generally 1:1 (ppm) (Radke et al. 2000; McLean 2003), which is reinforced by the position of the regional GAB samples in Figure 3a. The Namoi River and other regional streams have lower $Na^+$ and $HCO_3^-$ concentrations and a lower $Na^+/HCO_3^-$ ratio than both the historic GAB data and the deeper alluvial data collected in this study.3a~~ shows a mixing line that the alluvial samples follow, plotting between the end-members of the GAB and the Namoi River, suggesting . This suggests that there is an increasing GAB contribution to the alluvial groundwater with depth. This also suggestsimplies and that a continuum of mixing exists between the shallow and deep groundwater within the LNA. The shallow samples (25220-1 and 30259-1) that are more $Na^+$ enriched compared to samples from the GAB have undergone separate evapotranspiration processes and hence have a concurrent increase in $Cl^-$. Assuming that $Cl^-$ is behaving conservatively (Appelo & Postma 2005) we surmise that increases in dissolved major ion concentrations concomitant with increases in $Cl^-$ in the shallow groundwater are likely to be a result of evaporation.

Further hydrochemical evidence for these recharge mechanisms is the covariation of $Na^+$ and $F^-$, both interpreted as primarily derived from groundwater interaction with silicate minerals in this region (Airey et al. 1978; Herczeg et al. 1991; McLean 2003) (Figure 3b). Our alluvial samples fall on the mixing line between samples from the river and nearby tributaries and regional samples from the GAB (Radke et al. 2000), in a similar way to the $Na-HCO_3$ trend that we observe in Figure 3a. The Cl/Br ratios in the groundwater also support the mixing interpretation provided by the $Na^+$ and $HCO_3^-$ concentrations, contrary to the possibility of water rock interactions along the alluvium flowpath (Figure 3c). Furthermore, the relationship between $^{36}Cl$ and $Na^+$ provides additional evidence of mixing in the groundwater (Supplementary Figure 1).

Figure 3 also showhighlights the aforementioned deep outlying sample (273314), which was 207 m bgs in total depth, yet plots with the shallow alluvial and river samples. Figure 2 shows that this sample is situated just above the Napperby Formation. We hypothesise that this sample originated from surface recharge from the Namoi River (which is in contact with the underlying Digby Formation to the south of the study area), with negligible input from the more $Na-HCO_3$-rich groundwater in the Pilliga Sandstone, where the sample is from.  sample 30345-2 (Supplementary Tables 2 and 3), which is situated in the lower part of the LNA in proximity to the alluvial contact with the Napperby Formation (Figure 2) has a similar geochemistry. These results suggest the connection between deeper Triassic formations beneath the GAB and the Namoi River, which must be an important consideration in future water balance models of the catchment.

~~The evapotranspiration process is also shifting the groundwater composition towards calcite saturation. Both processes contribute to increasing the $Na^+$/$HCO_3^-$ ratio. The evaporative enrichment is also evident in the concentration of $F^-$, $Cl^-$ and the $Cl^-$/$Br^-$ ratio (Figure 4). Evidence for the $CaCO_3$ precipitation is found in the calcrete material on the surface soils, which also occurs in other semi arid environments due to this process (Meredith et al. 2016).~~

separate mechanisms of recharge; surface water recharge plotting along an evaporation trend line and potential inflow from the GAB clustered with regional samples from the GAB (McLean 2003).

The $\delta^{18}$O and $\delta^2$H compositions suggest two mechanisms of recharge to the alluvium (Figure 3b; Supplementary Table 3): artesian discharge and surface water infiltration. The regional GAB samples ($\delta^{18}$O and $\delta^2$H: -6.58‰ to -6.24‰ and -43.1‰ to -38.8‰, respectively (McLean 2003)) plotted within the alluvial groundwater sample range ($\delta^{18}$O and

$\delta^2$H: -7‰ to -6‰ and -44‰ to -37‰, respectively). This may suggests a GAB component in the alluvium. A second trend is observed with alluvial groundwater samples ranging from -

3.4‰ to -1.4‰ for $\delta^{18}$O and -20.5‰ to -11.5‰ for $\delta^2$H plotting along an evaporation trend line that suggests good connection and mixing between modern surface water and shallow groundwater.

Further hydrochemical evidence for these recharge mechanisms come from assessing the covariation of $Na^+$ and $F^-$, both interpreted as primarily derived from groundwater interaction with silicate minerals in this region (Airey et al. 1978; Herczeg et al. 1991;

McLean 2003) (Figure 4a). Our alluvial samples fall on the mixing line between samples from the river and nearby tributaries and regional samples from the GAB (Radke et al. 2000), in a similar way to the Na HCO$_3$ trend that we observe in Figure 3a. The Cl$^-$/Br$^-$ ratios in the groundwater also support the mixing interpretation provided by the $Na^+$ and HCO$_3^-$

concentrations, contrary to the possibility of water rock interactions along the alluvium flowpath (Figure 4b). The Cl/Br ratios in shallow samples connected to the river are consistent with expected ratios in rainfall (Short et al. 2017). The regional GAB samples (Radke et al. 2000) show a Cl$^-$/Br$^-$ ratio closer more similar to seawater, with our samples from the LNA lying on a mixing trend between the two end-members.

[Figure]

**Figure 4.** a) Na⁺ vs F⁻ and b) Cl⁻/Br⁻ vs Cl⁻, highlighting the mixing trend between the surface recharge and the GAB that we observe in other geochemical indicators. The red dotted line represents the Cl⁻/Br⁻ ratio for rainfall.

Figure 4a also reveals a deep outlying sample (273314), which was 207 m bgs in total depth (screened 182 195 m bgs), yet plots with the shallow alluvial and river samples. Additionally, many of the geochemical parameters (for example Na⁺, F⁻, HCO₃) in this sample have a signature more similar to river water rather than what would be expected in the GAB 207 m bgs (Supplementary Tables 2 and 3). Figure 2 shows that this sample is situated just above the Napperby Formation. This suggests that this sample originated from surface recharge from the Namoi River (which is in contact with the underlying Digby Formation to the south of the study area), with negligible input from the more Na HCO₃-rich groundwater in the overlying Pilliga Sandstone. We observe a similar geochemistry in sample 30345-2 (Supplementary Tables 2 and 3), which is situated in the lower part of the LNA in proximity to the alluvial contact with the Napperby Formation (Figure 2). These results suggest the connection between deeper Triassic formations beneath the GAB and the Namoi River, which must be an important consideration in future water balance models of the catchment.

*5.1.1 Mixing between groundwaters of varying residence times*

Major ion and water stable isotope data suggest two primary mechanisms of recharge to the LNA and show that mixing is occurring within the alluvium.  that there is  $^3$H activity and $^{14}$C content in the alluvial groundwater to quantify the potential residence times of the groundwater sources that are mixing within the alluvium. Tritium activities above the detection limit at depth (down to 207 m bgs) indicates the extent of recharge from episodic flooding. Measuring $^3$H above the detection limit at these depths also shows that surface recharge reaches the deeper LNA relatively quickly (< 70 years). Past $^3$H data from the region (Calf 1978) suggest that $^3$H was already present in the deeper parts of the alluvial aquifer (> 70 m bgs) prior to a major flood in 1971, with activities ranging from 7.9 TU to 11.2 TU. This indicates good connectivity to the surface. Additionally, measurements of $^3$H post-flooding (16.6 to 20.7 TU) indicate that substantial recharge took place during this flood, highlighting the importance of surface water recharge to the LNA.  $^{36}$Cl,

 $^3$H  $^3$H ~~remains relatively prevalent in the deeper part of the system. This indicates the extent of recharge from episodic flooding and shows that surface recharge reaches the deeper LNA (down to ~80 m bgs) relatively quickly (< 70 years). In February 1971, the region experienced the second largest flood on record. Pre-flood sampling of deep groundwater (> 70 m bgs) revealedactivities ranging from 7.9 to~~

[3]

[3]

[3]

[revised manuscript text omitted]

**5.5 Implications for sustainable groundwater use**

The continued sustainable access to groundwater is vital for irrigation, stock and domestic water supplies in the study area. Increased reliance by the irrigation industry on GAB groundwater with high $Na^+$ concentrations and very long residence times could have negative environmental impacts, such as producing sodic soils, as well as a significant economic impact. The difficulty in accurately constraining how the artesian contribution to the LNA will change over time means consistent monitoring of the groundwater is important for assessing changes to groundwater quality and quantity and the impact that this will have on the irrigation industry in the region. Additionally, the percentage extent of the interaction between the GAB and the LNA (Figure 8), and how this percentage changes over time depending on surface water recharge and increased groundwater extraction, has repercussions for the continued access and management of groundwater in the LNA. In regions where very old groundwater is used, assessments of sustainability must consider changing water quality (for example salinity and the sodium adsorption ratio (SAR)), as well as changes to

[revised manuscript text omitted]

---

## Author Response (AR2)

**This is a much better paper than the original version and I consider that it is suitable for publication in HESS following moderate revisions. I have made several specific comments below that are mainly concerned with the details of the paper. The overall methodology and conclusions are sound and the aims and objectives are generally clear.**

We thank the reviewer for taking the time to go through our manuscript again. We have addressed all specific comments below. All line numbers below refer to the track changed document.

**The one major concern that I have is with the calculation of 36Cl residence times. The interpretation of 36Cl is difficult in the best of circumstances due to problems with defining the 36Cl input function over time (given that the rainfall patterns, distance from the coast, input of dust etc is likely very different several hundred thousand years ago to today). Additionally, in many studies (including this one), the in situ production rate of 36Cl is assumed rather than measured and the definition of residence times via Eq. (1) implicitly assumes piston flow which is also unlikely. In groundwater such as that discussed in this study, there is also the problem of mixing which means than the calculations can only be carried out in waters that can be guaranteed to be the pre-mixing endmembers (which given the extent of mixing may be difficult). The fact that the GAB groundwater has long (several 100 ka) residence times is well established in previous studies and this paper does not add to that understanding. It would be sufficient to note that the low 36Cl activities are consistent with the old GAB groundwater and focus on the mixing.**

We agree about the shortcomings of 36Cl/Cl to calculate meaningful groundwater residence times and we do make these shortcomings clear in the introduction (lines 95-102). Previous literature in the GAB (Bentley et al. 1986; Radke et al. 2000; Love et al. 2000; Moya et al. 2016) and the Coonamble Embayment (Radke et al. 2000; Mahara et al. 2007) have constrained GAB groundwater residence times using similar or the same methods (this is outlined at lines 408-413 in the revised manuscript). Therefore, the 'old' nature of GAB groundwater has been established in most cases with the same methods and uncertainties. However, we have followed the reviewer's advice above and removed all references to residence times associated with 36Cl from the discussion.

All references to 36Cl in the results section are purely the 36Cl results from our samples so that section remains (lines 535-540).

The initial discussion regarding 36Cl (636-645) just discusses Figure 7 and the distinct mixing trend between the samples so that remains.

At line 648 is where we first mention residence time calculations using 36Cl. These lines have been deleted.

Lines 654-667 remain because they are just discussing the 36Cl values in relation to the 14C values that we obtained for our samples.

Lines 674-680 remove all trace of residence time estimation. At line 676 "up to 900 ka" has been removed and replaced with "conceivably older than that of the GAB recharge zone".

"…groundwater with a residence time of 900 ka (calculated using eqn 1.)" has been changed to "…groundwater with low 36Cl activities, consistent with old GAB groundwater" (lines 679-680). The final sentence (line 680) regarding how the "only source of groundwater with a residence time of 900 ka in the study area is the GAB" has been deleted, as this is now evident in the previous sentence.

Lines 817-818: mention of residence time has been removed and 900 ka has been replaced with "old groundwater consistent with the GAB".

Lines 39-40 in the abstract have removed reference to 900 ka and instead mention groundwater that is potentially hundreds of thousands of years old, which is consistent with that of the GAB.

The residence times have also been removed from Figure 7 and replaced with 'alluvial' and 'GAB'.

These are all the instances where we refer to residence times calculated from 36Cl. They have all been removed and instead the low 36Cl data from the LNA has been used to compare with the GAB.

**Specific Comments**

**Lines 50-55. Strictly recharge is infiltration of rainfall, what you are discussing here is mixing between recently recharged waters and older upward flowing groundwater. Since your main story is mixing and the water balance then I would frame the first sentences around that topic.**

The first sentence of the introduction has been changed to focus on mixing rather than recharge. It has been changed to "Groundwater type mixing in an alluvial aquifer can occur between recently recharged groundwater (via infiltration from the land surface) and groundwater discharging into the alluvium from surrounding geological…" (lines 53-55 in the revised manuscript). Additionally, lines 58-59 have been modified to reflect mixing rather than recharge. "Additional uncertainties when allocating recharge to each source include…" has been changed to "Additional uncertainties confounding source attribution include…" (lines 60-61).

**Lines 62-75. Similar comments apply. The Scanlon study is specific to recharge but some of the others deal with mixing in aquifers.**

At line 68 we have included "…discharge from…" when mentioning artesian sources, to make the distinction between recharge and discharge. At line 74 we have changed "..can improve our understanding of recharge processes…" to "…can improve our understanding of groundwater mixing processes…". At lines 84-85 we have changed "…proportioning sources of recharge to groundwater resources" to "…proportioning input sources for groundwater that has mixed origins".

**Lines 79-83. Not very clearly expressed. It would be good to reference Jascheko (2016, Chemical Geology, 427, 35-42) who deals with this in some detail.**

This has been better expressed at lines 90-93 in the revised manuscript. We have deleted the sentence regarding different tracers for a given window of time and included a sentence on the hindrance that calculation assumptions can have on interpretations. (line 90-91). We have cited the above reference at this location. We now mention that "multiple tracers are **useful for covering** the relevant time scales **and uncertainties associated with** the large range of groundwater residence times" (lines 92-93) (new text in bold).

**Lines 89-93. The interpretation of 36Cl is also hampered by uncertainties in the input function. Because rainfall R36Cl values vary temporally due to climate variations the input function at any time in the past may not be the same as it is today (e.g., Phillips et al., 2000. In: Cook & Herczeg, Environmental Tracers in Subsurface Hydrology. pp. 299-348).**

This additional difficulty in the interpretation of 36Cl (regarding the input function and changing rainfall) and reference has been added at lines 101-104. However, this is the case for all pre-existing 36Cl/Cl work in the GAB. Whether climatic variations are more important than groundwater flow assumptions, neotectonic movements or even point source contributions from high U/Th concentration rocks, is a matter of conjecture. We provide a dataset and interpretations that can be compared to other work in the area.

**The introduction could be clearer and framed more specifically around understanding mixing between groundwater from different sources in aquifers. As it is you use recharge and discharge alternately to describe the input of the deeper waters in the alluvials. Also it would be good to note that geochemistry allows us to understand long-term patterns of mixing and groundwater flow whereas using hydraulic heads (especially in systems perturbed by land clearing or water abstraction) only informs present day groundwater flow.**

We have changed the introduction to better reflect a focus on mixing rather than recharge (as per the first two comments above). We have carefully gone through the introduction and changed any mention of input of artesian water to 'discharge' rather than recharge, so that there is consistency (this includes at lines 68, 73, and 87).

Lines 126-139 (in the revised manuscript) have been changed to reflect the final sentence in the review comment. Line 130 is now a new paragraph (forming the concluding paragraph of the introduction). Lines 126-129 have been added, as per the reviewer's suggestion, to note that geochemical data can give us insights into long term patterns and trends, whereas hydrologic data (such as hydraulic heads) can give us insights on seasonal pumping impacts, and current local and catchment-scale groundwater flow paths.

**While the choice of references is always personal, there has been more done in this field than is apparent from the introduction. For example, the numerous references by Edmunds (especially the review in Applied Geochemistry, 24, 1058-1073). These are also several Australian examples of the use of major ions, stable isotopes, and radioactive isotopes by CSIRO (Herczeg and co-workers) and the Monash (Cartwright and co-workers) groups.**

We have included Edmunds (2009) in the references in the introduction, when highlighting the importance of geochemistry for other groundwater processes (line 76 in the revised manuscript). References from Herczeg et al. 1991 and Cartwright et al. 2010, 2013 are also included throughout the manuscript (lines 466, 567, 622, 97).

**Lines 109-115. This is a rather convoluted way to say that you use the radioisotopes to determine mixing between waters with long and short residence times. It is obvious that determining residence times in mixed waters is nigh on impossible and I don't think that you need this as your explanation above is clear.**

We have removed lines 109-113. We have left the last sentence to illustrate the global relevance of our study (lines 137-139 in the revised manuscript).

**Line 134 (and elsewhere) ML is a commonly used unit in Australia but less so elsewhere. Suggest using m3 (which is the conventional SI unit).**

We have changed all ML to m3. (lines 169 and 174 and 249-253).

**Section 2 is much more relevant and informative than in the previous version of the paper. Adding the groundwater flow directions to one of the maps and the cross-sections would be good.**

The major direction of groundwater flow actually follows the B-B' cross section line in Figure 2. In the interest of not making the maps too busy, we have mentioned in the caption that the major flow direction is from the SE to the NW following the B-B' line.

**Lines 245-246. Can you give typical screen intervals (it is important as interpretation of geochemistry from short-screened wells is much easier than from those with long screens)?**

We have provided the average length of the screened interval across all samples (average length of 5.6 m) at line 295-296. The specific depth of the start and end of all screen intervals are detailed in the tables in the supplementary.

**Lines 268-275. A bit of a longwinded way of saying that you used existing data to characterise the GAB groundwater.**

We have removed the lines stating our intentions during the sampling campaign (at line 319 in revised manuscript) to shorten this paragraph. We now simply state that we couldn't access any GAB bores so we used past data. We have also cut some words at lines 320 and 336 to condense this section.

**Lines 278-283. This point about the aquifer structure was made earlier. While it might be of local interest, I am not sure that it is worth repeating through the paper (and in many ways it is misplaced as you are stating a conclusion in the methods section). It is a common observation globally that geological formations (which might be determined on ages) do not always form separate aquifers. Your explanation on lines 178-183 explains it well enough.**

We have removed this section (removed at line 340 in revised manuscript) as we have already stated it previously and is misplaced in the methods.

**Lines 290-291. That is precision not accuracy.**

We have changed 'results are accurate to' to 'results have a precision of' at lines 347-348 in the revised manuscript.

**Lines 335-339. You seem to be describing the procedure for getting the CBE correct. If you determined NH4 and the subsequent CBE was OK, then just make sure that the NH4 concentrations are in the table and report your final CBEs.**

We thank the reviewer for noting this detail that we have overlooked many times, and we apologise for not finding it and remedying it earlier. We did a qualitative field analysis, which suggested the presence of NH4, however we were unable to properly measure it along with the other ions. Therefore, to avoid confusion, we have eliminated its mention and instead reported the outlying charge balances (-7.8 % for sample 25327-1 and -5.8 % for sample 36001-1). This is at line 404 in the revised manuscript.

**Lines 342-348. These details are getting beyond strict methodology and it would be better to discuss the exact compositions of the endmembers when you present the results of the calculations.**

We have deleted these lines at 408 in the new manuscript from the methods. The lines detailing the general Cl concentration of surface water and GAB groundwater have been moved down to lines 689-691 in the new manuscript (in the discussion section). This then flows onto the discussion regarding the Cl mixing in our study.

**Lines 349-359. This section is fine. However, as noted elsewhere these calculations do not add much to the paper and are highly speculative given that mixing has occurred.**

We have modified this section to state that we calculate residence time estimations from 36Cl to allow a direct comparison, under similar assumptions, with other estimates obtained from

GAB groundwater elsewhere. Even though we don't explicitly put a residence time on the groundwater using the 36Cl anymore, we still think it is important to show that some of our alluvial samples were in the same calculation range as other GAB groundwater and that the methods we used are the same for residence times calculations for GAB groundwater in the literature (lines 409-441 in the revised manuscript).

**Lines 363-403. This section describing the geochemistry is far better than in the first version of the paper. The only thing that I would suggest adding is the standard deviation to the average concentrations.**

We have included standard deviations to all average concentrations in this section (lines 446-461 in new manuscript).

**Fig. 3a. It is not clear what the calcite saturation line is (explain in caption or text).**

We have added to the caption for Figure 3a: "The orange calcite saturation line indicates samples that are more enriched due to separate evapotranspiration and calcite precipitation".

**Lines 406-414 (and elsewhere). Given the quoted precision your δ2H values should not have decimal places and δ18O values should only have 1 decimal place.**

Any future use of this research would be compromised by an artificially introduced rounding error. For example, rounding up δ2H from -10.4 to -10 and δ18O from -3.86 to -3.9 would change a d-excess calculation by approximately 1‰. It is common practice to report one decimal figure for δ2H and two decimal figures for δ18O for the precisions reported. Furthermore, in some instances stable isotope runs have even better precision on individual runs than reported but a general long term precision is reported.

**Lines 410-412. Add a reference for the evaporation trend and perhaps rephrase it as "define a trend to the right of the meteoric water line with a slope of XX that is consistent with evaporation (REF)".**

We have deleted "follow an evaporation line" and replaced it with the reviewer's suggestion (lines 495-496 in the revised manuscript). We have included a reference to show that this is consistent with evaporation (Cendón et al. 2014). This reference shows a water stable isotope plot with a groundwater evaporation line trending to the right of the meteoric water line.

Cendón, D.I. et al. Groundwater residence time in a dissected and weathered sandstone plateau: Kulnura-Mangrove Mountain aquifer, NSW, Australia. *Aus. J. Earth Sci.* **61(3)**, 475-499, 2014.

**Lines 409-410. Are the local sites close? Given that your samples seem to lie on the GMWL, I'm not sure whether the LMWLs are relevant (?)**

Gunnedah is 86 km away from the study site and the Macquarie Marshes are 210 km away. Whilst they are not extremely close, considering the low relief orography in Australia, they are relevant. They are the closest LMWL's for this study site and so we believe they are important to put our data into regional context. Therefore, we have left the mention to them in text (line 493 in revised manuscript).

**Lines 422-433. Here and elsewhere I would specify "<0.04 TU" rather than "detection limit" in the text as it is more specific.**

We have changed "above the det limit" or "below the det limit" to > 0.04 TU and < 0.04 TU at all locations where it was mentioned (including lines 516, 593-594, 602, 608 in the revised manuscript).

However, we were imprecise with our words when we referred to the 0.04 TU threshold as a 'detection limit'. It is actually a quantification limit, meaning that we can still detect values below 0.04 TU, however there is too much uncertainty to use them confidently. We set the limit at 50 % uncertainty and so if the analysis uncertainty is 50% or higher, then we say it is below the quantification limit. Keeping the values that were below the quantification limit allows us to plot our data against other isotopes like 14C.

Any mention of 'detection' has been changed in text as per reviewer's suggestion above, or has been changed to 'quantification' (in Figures 5 and 6).

**Lines 423-425. The statement that you have modern recharge is an interpretation. Try to rephrase it so you explain to the reader why you said this (eg "The highest 3H activities of XX are near the river… these are similar to those of modern rainfall(?)… this implies…"). On that point I presume that you can infer a modern rainfall 3H activity from the Tadros (2014) compilation and that would be worth doing.**

We have deleted this sentence as it was and changed it to: "The highest $^3$H activities of 2.31 TU and 2.36 TU are from a sample 40 m from the river and from the Namoi River itself, respectively. These are very similar to modern rainfall in Australia (2-3 TU (Tadros et al. 2014)), which suggests modern recharge near the river channels" (lines 511-515). Tadros et al. 2014 has been added to the reference list as well.

**Lines 426-429. This is true but the explanation is a little brief for anyone not familiar with southern hemisphere 3H. A couple more sentences would be helpful.**

We have added a qualifier saying "…the peak of the bomb pulse in Australia was only 60 TU**, compared to locations in the northern hemisphere. This is primarily because most thermonuclear testing was undertaken in the northern hemisphere far from Australia and mixing is limited between the atmospheric convection cells in the northern and southern hemispheres.** Therefore, 3H in Australian rainfall has been at background…." (new text in bold; lines 517-521 in revised manuscript). This places the 3H Australian concentrations into context for international readers.

**Fig. 5. Suggest plotting the samples that are below detection on the left hand axis as the 3H values have no actual meaning. The same is true in your tables (ie designate these as "<0.04" or "bd" rather than quoting actual values).**

Please see our response above regarding quantification limits. The legend in figures 5 and 6 has been changed to state 'quantification limit' rather than 'detection'.

**Lines 492 to 497. This section is not very clearly expressed. You could add a few more details (eg where in the region these observations come from and whether the data are from the same wells). Also you should stress that both observations (the high initial 3H and the subsequent increase in 3H) imply recharge from the surface.**

This has been rewritten to better express what we are trying to say (lines 596-602 in the revised manuscript). We have changed the beginning to "Tritium data from the 1970's collected from bores that were included in our sampling campaign (25329 and 25332) (Calf 1978) suggest…" We have also emphasised that both observations imply recharge from the surface and noted this at lines 600-602.

**Line 492. Where does the estimate of <70 years come from?**

This estimate at lines 595-596 comes from the half-life of 3H. We are saying that if we can measure 3H > 0.04 TU at depth (207 m bgs), then 3H has not undergone decay longer than 70 years at this stage. Hence, if we can detect 3H at depth it must have been recharged in a timeframe of < 70 years.

**Lines 503 to 505. Does this interpretation agree with the major ion geochemistry? For example if your deeper water is 14C-free and the surface water has a 14C activity close to 100 pMC, then given the relative HCO3 concentrations how much old water do you need to add to reduce the 14C down to the observed values? What then would the predicted 3H activities of the mixed water be? The interpretation is plausible but it would be good to see it confirmed.**

We thank the reviewer for these comments and ideas. This suggestion is looking for complementary mixing indicators, which would reinforce the concept already illustrated by the Cl mixing calculations and with the 36Cl vs 14C plot, and even the 36Cl/Cl vs Na plot that is included in the supplementary. We feel that the requested additional calculations are beyond the scope of the manuscript and do not provide critical insights. The reviewer's suggestions could form the basis of material for a new paper.

**Fig. 6. I found the reversed axis for 14C confusing (is there any reason why 0 is not at the left especially since 3H is plotted conventionally). Also due to the waters containing different DIC concentrations, mixing lines are curved (if you know the relative DIC**

**concentrations, you should be able to predict whether the curvature is concave or convex).**

Yes, we reversed the 14C axis to intuitively place samples with proximity to surface on the top left-hand corner. However, it was easy enough to reverse and we have done so to facilitate interpretation. The new Figure 6 contains the reversed 14C axis following reviewer advice. The blue dotted mixing line in Figure 6 was not a calculated mixing line, but rather simply a visual aid for readers, to highlight that there is this mixing between groundwaters. This is why the line isn't curved.

**Lines 531-562. As outlined above, I think that estimating residence times using 36Cl in waters that have undergone mixing is not very convincing. It is clear from the available literature that the GAB waters have residence times of several hundred thousand years and this study does not add to that understanding. I strongly suggest that this section is reframed to discuss the mixing with a statement referenced to previous works that the low 36Cl in GAB groundwater is a consequence of the (already established) long residence times.**

As detailed above, all mention of residence times calculated directly from our 36Cl values have been removed. Instead we now frame the discussion as mixing between very young groundwater and groundwater that is potentially hundreds of thousands of years old, which has 36Cl values consistent with those from the GAB (Bentley et al. 1986; Radke et al. 2000; Love et al. 2000; Mahara et al. 2009; Moya et al. 2016).

**Lines 546-547. The Cl/Br ratios are a better indication of the lack of halite dissolution than the Cl concentrations.**

Yes, we agree and this can be readily observed in Figure 3c, which also indicates expected Cl/Br ratios trends following dissolution of evaporites.

**Lines 593-597. As in the introduction, it is worth stressing here that the geochemistry provides an indication of the long-term groundwater flow and mixing whereas hydraulics in perturbed systems show how the system is currently functioning.**

Reference to the long-term insights from the geochemical data, contrasted to the current system functioning using hydraulic heads has been mentioned again at lines 727-729.

**Section 5.3. This section appears almost as an afterthought (it has its own introduction, data description, and interpretations) and is speculative. While the data is interesting they are not that well integrated with the main part of the paper. I suggest shortening the material and noting that the 14C contents in some of the groundwater has changed over time implying a dynamic system without trying to over interpret it. As it is, the data seems to have been collected without a specific hypothesis in mind (eg collecting**

**from an area before and after groundwater extraction was established) and so are not that easy to interpret with any confidence.**

We have shortened this section, including removing the detailed speculation of why each bore that had a different 14C activity may be experiencing different rates of GAB inflow. We have changed this section now to better reflect that it is important that temporal changes in the extent of groundwater mixing are considered to make judgements about how the system may change into the future or with increased groundwater abstraction. We maintain that the 14C data from our study and the past data may offer some preliminary insights into how the extent of mixing has changed, with a higher pmc now reflecting increased modern infiltration. However, the over interpretation in this section has been removed, as per the reviewer's suggestion. All specific changes between lines 746 and 784 are shown in the track changed document.

**Conclusions. These are a summary of the paper. Given HESS is an international journal, it would be good to emphasise what is of broader interest. Some of that material (changes over time, general utility of tracers) appears in Section 5 but probably would be better here to round off the study.**

We have removed a portion of the conclusion that was focused on the detailed findings of section 5.3 (and very site specific oriented as well), as the weight put on that section has now been greatly reduced. At lines 813-815 we include why out findings are important globally and lines 844-847 already has a concluding sentence that presents our findings with a broader interest. Since one of the major findings of this manuscript was the difference between water balance modelling estimates of GAB contribution and our findings using geochemistry, it is necessary to reference local sites in the conclusion. However, we have attempted to convey through the conclusion and the final paragraphs of the introduction too that water balance models, globally, must consider geochemistry because of the discrepancies that we find in our study and the ramifications for the sustainable use of groundwater.

[revised manuscript text omitted]